# Air pollution particles hijack peroxidasin to disrupt immunosurveillance and promote lung cancer

**Zhenzhen Wang[1,2], Ziyu Zhai[1], Chunyu Chen[1], Xuejiao Tian[1], Zhen Xing[1], Panfei Xing[2], Yushun Yang[1], Junfeng Zhang[1]\*, Chunming Wang[2]\*, Lei Dong[1,3]\***

[1]State Key Laboratory of Pharmaceutical Biotechnology, School of Life Sciences, Nanjing University, Nanjing, China; [2]State Key Laboratory of Quality Research in Chinese Medicine, Institute of Chinese Medical Sciences, University of Macau, Macau, China; [3]Chemistry and Biomedicine Innovative Center, Nanjing University, Nanjing, China

**Abstract** Although fine particulate matter (FPM) in air pollutants and tobacco smoke is recognized as a strong carcinogen and global threat to public health, its biological mechanism for inducing lung cancer remains unclear. Here, by investigating FPM's bioactivities in lung carcinoma mice models, we discover that these particles promote lung tumor progression by inducing aberrant thickening of tissue matrix and hampering migration of antitumor immunocytes. Upon inhalation into lung tissue, these FPM particles abundantly adsorb peroxidasin (PXDN) – an enzyme mediating type IV collagen (Col IV) crosslinking – onto their surface. The adsorbed PXDN exerts abnormally high activity to crosslink Col IV via increasing the formation of sulfilimine bonds at the NC1 domain, leading to an overly dense matrix in the lung tissue. This disordered structure decreases the mobility of cytotoxic CD8$^+$ T lymphocytes into the lung and consequently impairs the local immune surveillance, enabling the flourishing of nascent tumor cells. Meanwhile, inhibiting the activity of PXDN abolishes the tumor-promoting effect of FPM, indicating the key impact of aberrant PXDN activity on the tumorigenic process. In summary, our finding elucidates a new mechanism for FPM-induced lung tumorigenesis and identifies PXDN as a potential target for treatment or prevention of the FPM-relevant biological risks.

**\*For correspondence:**
jfzhang@nju.edu.cn (JZ);
cmwang@umac.mo (CW);
leidong@nju.edu.cn (LD)

**Competing interest:** The authors declare that no competing interests exist.

## Editor's evaluation

This article focused on the bioactivity of inhaled fine particulate matter (FPM) in promoting lung tumor progression. The authors presented carefully performed work with impressive quantity. They shed light on that FPM-accelerated tumorigenesis through disordering interstitial extracellular matrix in lung tissue and subsequently impairing early immune defense to tumor cells. Besides, they found that FPM's bioactivities are endowed by an unexpected enzyme, peroxidasin, related to the collagen crosslink, and the latter's abnormal high enzymatic activity. These findings are promising and provide a new potential target for preventing FPM-relevant diseases.

## Introduction

Inhalable fine particulate matter (FPM) with a diameter less than 1 μm in air pollutants and tobacco smoke has been recognized as a group 1 carcinogen and substantial threat to global health (*Lim et al., 2012*). About 10 μg/m³ increase in its concentration was correlated with an 8% rise in lung cancer mortality (*Guan et al., 2016*). However, its carcinogenic mechanism remains unclear. During

lung tumorigenesis, both the growth of cancer cells per se and the supporting microenvironment are crucial (*Martin et al., 2021*). Earlier studies propose that the particles directly induce gene mutations and carcinogenesis (*Wu et al., 2004*). Clinical data suggests smoking as the factor for the highest prevalence of somatic mutation among lung cancers (*Alexandrov et al., 2016*; *Alexandrov et al., 2013*). However, despite its mutagenic potential, recent investigations reveal that FPM does not directly promote (and even inhibit) the proliferation of lung cancer cells. These inconsistent findings suggest that FPM might have unidentified targets other than cancer cells in promoting tumorigenesis, such as immune cells that play key roles in tumor development (*Bissell and Hines, 2011*). Under normal circumstances, the immune system rapidly detects and suppresses the tumor progression at the initial stage (*Iwasaki et al., 2017*). Especially, the mostly 'informed' defender immunocyte – cytotoxic CD8⁺ T lymphocytes (CTLs) – protect against potential cancer through efficient migration and cytotoxic contact with transformed or tumorigenic cells that have emerged in the lung interstitial space (*DuPage et al., 2011*). Once this crucial immunosurveillance and defense process of CTLs were compromised, the tumorigenesis would be uncontrollable (*Joyce and Fearon, 2015*).

Under the chemotaxis of biochemical signals, the mobility of immunocytes depends on not only its intrinsic capacity but also the microstructure of the interstitial extracellular matrix (ECM), that is, the way paved for immune cells (*Krummel et al., 2016*). For the former, evidence about the direct effect of FPM on the immune cell's migration capacity was validated. It is estimated that tobacco smoke particulates (TSPs) could impair the migration function of macrophages to mycobacteria and lead to increased susceptibility to tuberculosis in smokers (*Berg et al., 2016*). For the latter, clinical evidence links a dense collagen matrix surrounding the tumor with the restriction of T cells' access (*Salmon et al., 2012*; *Valkenburg et al., 2018*). Based on these reports and analysis, we speculated that FPM could disturb the migration and distribution of T cells in lung tissue, thus impairing CLTs' immune defense capacity against cancerous cells and consequently promoting tumor progression.

To test this hypothesis, we set out to study the effect of FPM inhalation on CTLs' immune response and tumor development by using both transplantation (Lewis lung carcinoma [LLC]) and transgenic (*Kras^{G12D}Trp53^{-/-}*) mouse models of lung carcinoma (*Johnson et al., 2001*). First, we validated that FPM promotes tumorigenesis by impairing CTLs' migration towards cancerous cells. The defect was attributed to denser collagen structure induced by FPM on CTLs' migration path, generating the physical isolation around tumor cells. More interestingly, we found that FPM exerts this effect by adsorbing peroxidasin (PXDN) – a crucial enzyme specifically mediating collagen crosslinking at NC1 domain – and increasing this enzyme's activity to over-crosslink ECM and prevent CTLs migration, which eventually tolerates tumor progression.

## Results

### FPM promotes lung cancer development by hampering CTLs' migration

To analyze the effect of FPM on lung tumorigenesis, first, we collected and prepared particulate matter in air pollutants with diameter <1 μm (PM1) from seven locations in China and the tobacco smoke particle (TSP), respectively. Given that these particles displayed diverse morphology and physicochemical characteristic (*Figure 1—figure supplement 1*, *Figure 1—source data 1*), which is consistent with the material property of particles in other reports (*Kelly and Fussell, 2012*), we mixed PM1 with the equal proportion from each collection to eliminate the interference of sampling resources. Then, mice exposed to mixed PM1 (mixture) or TSP were analyzed on two cancer animal models: the syngeneic LLC inoculation model (LLC model) or the transgenic mouse model (*Kras^{G12D}Trp53^{-/-}*) as illustrated in *Figure 1—figure supplement 2A*. Gross view (*Figure 1—figure supplement 2B*) and histological hematoxylin and eosin (H&E) analysis (*Figure 1A*) indicated that the FPM treatment markedly increased the tumors' multiplicity and progression. As *Kras^{G12D}Trp53^{-/-}* mice could generate multifocal tumors corresponding to different grades of lung carcinoma (*Sayin et al., 2014*; *DuPage et al., 2009*), the histological grade of this model was further analyzed. Tumors in FPM-exposed lung tissue were mainly classified as grade II and the ones of grade III and IV were significantly higher, whereas the majority of tumors in the phosphate buffer saline (PBS) group were of grade I, showing FPM leads to more advanced lung tumorigenesis (*Figure 1B*). Statistical analysis suggested that the number of tumors in the FPM-treated group was significantly higher than that in the PBS group (about three- to fivefold higher in LLC model and twofold higher in *Kras^{G12D}Trp53^{-/-}* model) (*Figure 1C*). Furthermore,

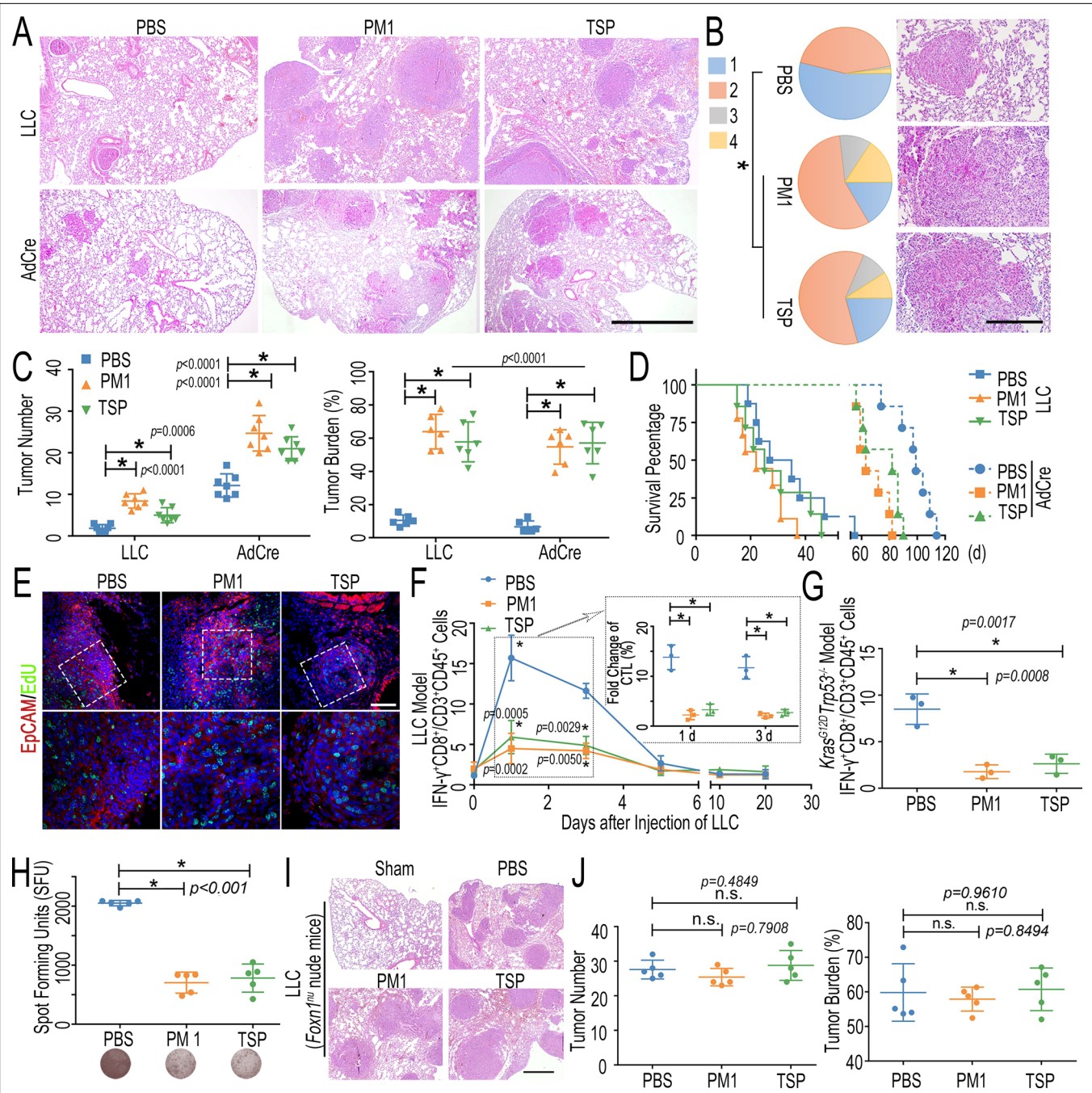

**Figure 1.** Fine particulate matter (FPM) accelerated lung tumorigenesis by inhibiting cytotoxic T cell lymphocyte (CTL) infiltration. (**A**) Representative hematoxylin and eosin (H&E) staining images of different model 20 or 50 days after FPM-exposed mice were stimulated with LLC or Cre-inducible adenovirus (AdCre). Scale bar = 100 µm. (**B**) Tumor stage (stages 1–4) of lungs in $Kras^{G12D}Trp53^{-/-}$ model (left) and representative tumors H&E staining images with higher magnification (right). p-Values are for comparisons of the percentage of stage 3 and 4 tumors in FPM-treated and control mice. n = 5. (**C**) Quantitative analysis of tumor number and tumor burden in lung tissue of different models treated as for panel (**A**). n = 5. (**D**) Survival analysis of mice exposed to FPM and subsequently stimulated with LLC or AdCre. n = 7. (**E**) Representative EdU staining images to analyze the proliferation rate of tumor site in lung tissue of LLC model 20 days after tumor initiation (DAPI, blue; EdU-positive cells, green). Scale bar = 100 µm. (**F**) The statistical analysis of CTLs (IFN-γ⁺CD8⁺/CD45⁺CD3⁺) in lung tissue based on flow cytometry at indicated day (0, 1, 3, 5, 10, and 20 days) after intravenous injection of LLC. n = 3. The inserted results in the dashed boxes indicate the fold change of CTLs in lung tissue 1 and 3 days after the LLC stimulation, relative to that under the physiological condition. (**G**) The statistical analysis of CTLs (IFN-γ⁺CD8⁺/CD45⁺CD3⁺) in lung tissue of $Kras^{G12D}Trp53^{-/-}$ mice based on flow cytometry 4 weeks after tumor initiation with the intratracheal injection of AdCre. n = 3. (**H**) IFN-γ enzyme-linked immunospot assay (ELISpot) in the lung tissue of OT-1 TCR transgenic mice 1 day after ovalbumin-Lewis lung carcinoma (OVA-LLC) stimulation (upper) and representative immunospot images (lower).

*Figure 1 continued on next page*

*Figure 1 continued*

n = 5. (**I**) Representative H&E staining images of lung tissue in *Foxn1^nu* nude mice exposed to FPM 20 days after intravenous injection of LLC. Scale bar = 100 μm. (**J**) Quantitative analysis of tumor number and tumor burden in lung tissue of *Foxn1^nu* nude mice treated as for panel (**I**). n = 5. Images are representative of three independent experiments. Results are shown as mean ± SD. *p<0.05 after ANOVA with Dunnett's tests.

The online version of this article includes the following source data and figure supplement(s) for figure 1:

**Figure supplement 1.** Morphology characterization of fine particulate matter collected from airborne pollution in different locations.

**Figure supplement 2.** The effect of fine particulate matter on tumorgenesis in two lung cancer models.

**Figure supplement 3.** Cell cytotoxic analysis of Lewis lung carcinoma (LLC) cells stimulated with a serial concentration of fine particulate matter (FPM) (0, 5, 10, 30, 50, 100, and 500 μg/mL) for 24 and 48 hr.

**Figure supplement 4.** The quantified analysis of EdU-positive cells in tumor site of lung tissue of Lewis lung carcinoma (LLC) model 20 days after tumor initiation.

**Figure supplement 5.** Flow cytometry analysis of CTLs in FPM-exposed lung tissue of mice after the LLC stimulation for indicated days.

**Figure supplement 6.** Immunofluorescence analysis for the CTLs in FPM-exposed lung tissue after the LLC stimulation for indicated days.

**Figure supplement 7.** Representative flow cytometry analysis of cytotoxic T lymphocytes (CTLs) (IFN-γ$^+$CD8$^+$/CD45$^+$CD3$^+$) in fine particulate matter (FPM)-exposed lung tissue of *Kras^{G12D}Trp53^{-/-}* mice 4 weeks after tumor initiation with the intratracheal injection of AdCre.

**Figure supplement 8.** Representative flow cytometry analysis of cytotoxic T lymphocytes (CTLs) (IFN-γ$^+$CD8$^+$/CD45$^+$CD3$^+$) in fine particulate matter (FPM)-exposed lung tissue of OT-1 TCR transgenic mice 1 day after ovalbumin-Lewis lung carcinoma (OVA-LLC) stimulation.

**Figure supplement 9.** Gross lung tissue images in fine particulate matter (FPM)-exposed *Foxn1^nu* nude mice 20 days after they were intravenously injected with Lewis lung carcinoma (LLC).

**Figure supplement 10.** Flow cytometry analysis of CTLs in FPM-exposed lung tissue of mice after the stimulation of chemokine IP-10 for 2 hr.

**Source data 1.** Physicochemical characteristic of fine particulate matter collected from airborne pollution in seven different locations.

**Source data 2.** Excel spreadsheet source file for *Figure 1C, D, F, G, H and J*.

the scenario was further validated by the corresponding tumor burden, based on the percentage of the area of tumor regions versus that of the total lung (about 7-fold more in LLC model and 10-fold more in *Kras^{G12D}Trp53^{-/-}* model). Moreover, in both models, PM1 and TSP exposure significantly shortened the survival of mice (*Figure 1D*). These results validated the correlation between FPM exposure and lung cancer development, in agreement with the epidemiological studies (*Atkinson et al., 2014*).

Next, we explored the reason for FPM promoting tumorigenesis. The conditions of the seeds and soil – the uncontrollably proliferative cancer cells and a tolerable immune microenvironment – are both crucial for tumor development (*Altorki et al., 2019*). We analyzed the effect of FPM on the tumor cells and their congenial microenvironment, respectively. Interestingly, FPM hardly promoted the growth of tumor cells and even inhibited their proliferation at higher concentrations (*Figure 1—figure supplement 3*). EdU incorporation assay was further employed to determine the impact of FPM exposure on tumor cells' proliferation in vivo. The result showed that the tumor site displayed similar replication capacity regardless of its size and advancement in these groups (*Figure 1E*, *Figure 1—figure supplement 4*), casting the doubt on FPM's direct promotion of tumor growth. These results inspired us to assess the effect of FPM on the immune microenvironment. Among these immunocytes related to immune surveillance, cytotoxic T lymphocytes (CTLs) as the most "informed" defender are critical for locally extinguishing the nascent tumor. The efficacy of these cells determines the fate of transformed cells – to death or flourish. Thus, we examined the change of CTLs' response in different groups during tumor progression (*Figure 1—figure supplement 5*). In the LLC model, CTLs in the PBS group were efficiently recruited into lung tissue to defend LLC stimulation, increasing up to about ninefold than that under the physiological condition at the initial stage (1–3 days). Conversely, the lung tissue with FPM exposure displayed blunt and insufficient early immune defense, with slight CTLs infiltration, decreasing by more than 60% relative to that of the PBS group, though there was no dramatic difference in CTLs accumulation among these groups at late stage (5–20 days). The immunofluorescence (IF) images also showed that FPM-exposed lung tissue was infiltrated with lower CTLs at the initial stage (*Figure 1—figure supplement 6*). These results indicated that the CTL's early immune response might not be normal in FPM-exposed mice and be decisive for the lung tumorigenesis, which is consistent with the reports in transgenic autochthonous lung tumors (*DuPage et al., 2011*). Then, we detected the CTLs' infiltration in *Kras^{G12D}Trp53^{-/-}* model 4 weeks after tumor initiation, during which the immune

response was reported to reach to the peak (*DuPage et al., 2011*). The results displayed similarly insufficient CTLs' defense in the FPM-exposed group (*Figure 1G*, *Figure 1—figure supplement 7*).

To further determine whether CTLs' reaction was specific to tumor cells, we further evaluated $CD8^+$ T cell activation in an OT-1 TCR transgenic mouse model, in which the $CD8^+$ T cells express a T cell receptor recognizing the SIINFEKL peptide of ovalbumin (OVA) (*Wang et al., 2020*). Upon the stimulation of OVA-expressing LLC (OVA-LLC) cells, the flow cytometry analysis of activated CTLs in lung tissue showed consistent tendency (*Figure 1—figure supplement 8*). OVA-LLC-specific immunity response in lung tissue was also tested by interferon-gamma (IFN-γ) enzyme-linked immunospot assay (ELISpot) (*Figure 1H*). The result further demonstrated that the antigen-specific-activated CTLs were significantly impaired by FPM exposure, decreasing to about 25% of that in the PBS group. Furthermore, to testify the indispensable role of CTLs on the tumor development, LLC cells were intravenously injected into FPM-exposed athymic $Foxn1^{nu}$ nude mice with T cell deficiency. As expected, the difference of lung tumorigenesis in PBS and FPM-exposed groups was abolished (*Figure 1I and J*, *Figure 1—figure supplement 9*), highlighting that the influences of FPM to tumor development were mediated by the CTLs. Additionally, without using tumor cells, we treated mice with a T cell chemokine – C-X-C motif chemokine ligand 10 (CXCL10) (*Griffith et al., 2014*), or named as interferon-inducible protein-10 (IP-10) – intrabronchially for 2 hr, as illustrated in *Figure 1—figure supplement 10A*, and found that the proportion of CTLs in FPM-treated mice (about 8%) significantly decreased compared with that of the PBS group (about 18%; *Figure 1—figure supplement 10B*). This finding is consistent with the scenario observed in the tumor (LLC and $Kras^{G12D}Trp53^{-/-}$) model and strengthens the conclusion that FPM exposure delays the CTLs' instantaneous defense response. The above results, taken together, indicate that FPM accelerates lung tumorigenesis via impairing CTLs' infiltration into the lung tissue.

## FPM hinders CTLs' migration by crosslinking type IV collagen and thickening tissue matrix

Next, we explored the reason for the impaired early response of CTLs under FPM exposure. CTLs' distribution in the lung interstitial tissue depends on its migration ability (*Weninger et al., 2014*), which is related to both the intrinsic activity of cells and the structure of the interstitial space formed by the local ECM on its migrating path (*Mrass et al., 2010*). We analyzed which factor was mainly affected by FPM. First, T cells treated with FPM in vitro or the CTLs separated from FPM-exposed lung tissue were respectively analyzed. Integrin-1 (ITGB-1), C-X-C motif chemokine receptor 3 (CXCR 3), and Rho-associated kinase (ROCKi) (*Overstreet et al., 2013*; *Tharaux et al., 2003*; *Ariotti et al., 2015*), the biomarkers related to CTLs' migration, were detected with quantitative real-time polymerase chain reaction (qRT-PCR). These results showed that FPM stimulation had little effect on the migration potential of CTLs (*Figure 2—figure supplement 1*). Second, we analyzed the change of the lung tissue structure after FPM exposure for 7 days. From scanning electron microscope (SEM images and quantitative analysis of the pore size of interstitial matrix) (*Figure 2A*, *Figure 2—figure supplement 2*), we noticed that the FPM exposure dramatically compressed the structure and crushed the interstitial space of the lung tissue. Further, Masson's trichrome staining indicated a higher density of collagen (*Figure 2B*). These data implied that FPM inhaled into the lung tissue condensed the native framework of ECM, which could block the path of CTLs migrating to the tumor site.

Consequently, we investigated in greater detail the movement of CTLs in an ex vivo model. The migration of CTLs in the slice of lung tissue (native or FPM-exposed) was analyzed by dynamically visualizing the cells' movement (*Figure 2—figure supplement 3*). According to the outcomes from time-lapse sequential images and trajectory analysis of CTLs' migration (*Figure 2C*, *Figure 2—videos 1–3*), in the lung tissue exposed to FPM, CTLs struggled to migrate, while those of the PBS group displayed quick migration pattern. Statistical analysis further validated that compared with the PBS group the FPM-treated lung tissue severely hindered the migration of CTLs (*Figure 2D*), which were weakly motile and showed insufficient displacement, distance, and velocity (*Bougherara et al., 2015*). Therefore, the change in the interstitial space, rather than attenuated migrating potential of CTLs per se, is responsible for the weakened infiltration of these cells in the FPM-treated lung tissue.

Then, we asked what caused the change of the lung structure after FPM exposure. Collagens, the main ECM components (*Gaggar and Weathington, 2016*), especially three kinds of ones enriched in lung tissue, including the type I, III, and IV ones (Col I, Col III, and Col IV) were focused on *Laurent,*

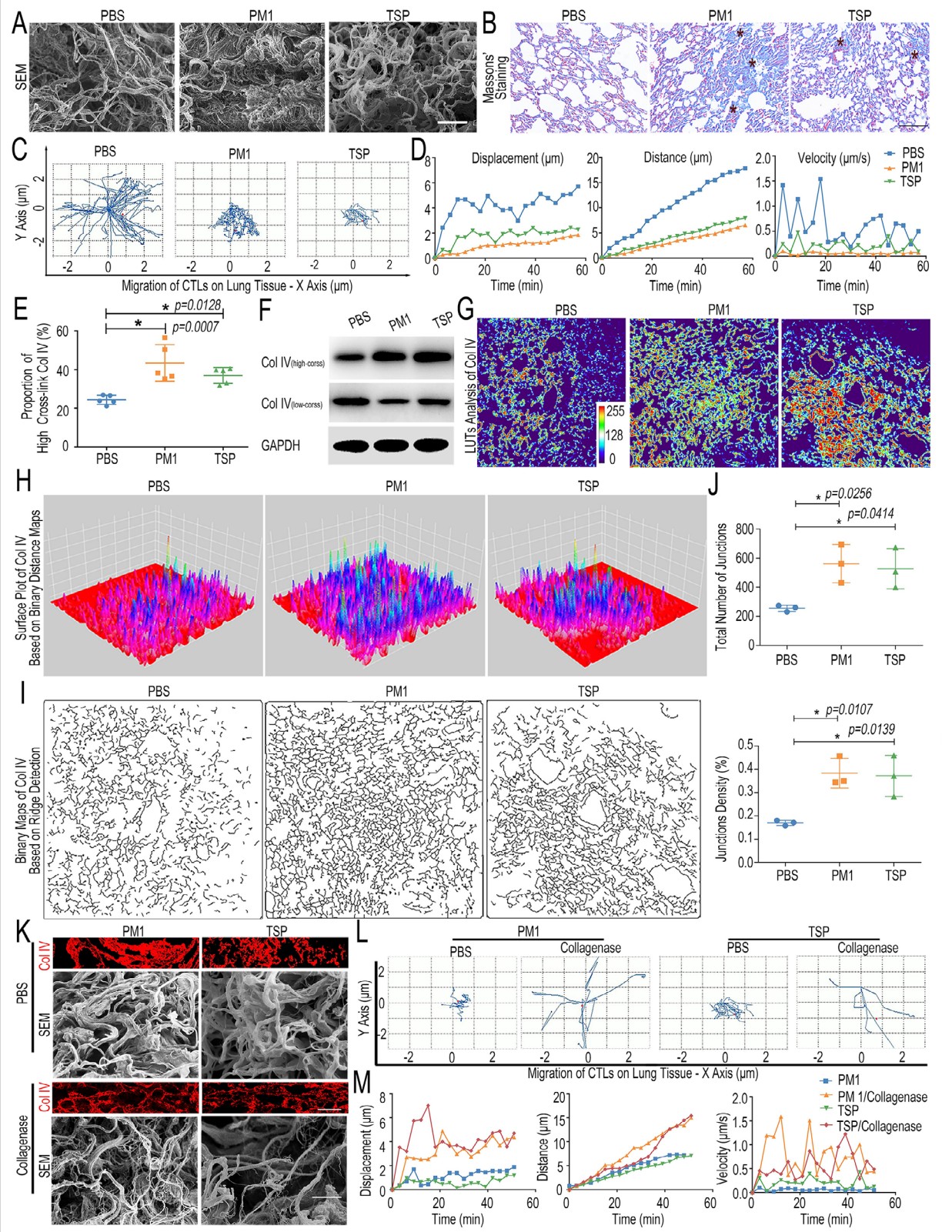

**Figure 2.** Fine particulate matter (FPM) impaired cytotoxic T lymphocytes' (CTLs') migration by increasing Col IV crosslinking in the lung tissue. (**A**) Scanning electronic microscope (SEM) images of the interstitial matrix in the lung tissue exposed to FPM for 7 days. Scale bar = 100 μm. (**B**) Representative Masson's trichrome histological analysis of lung tissue exposed to FPM for 7 days. Images are representative of three independent experiments. Scale bar = 100 μm. (**C**) Representative trajectory of CTLs' migration in lung tissue slice of FPM-exposed mice or phosphate buffer

*Figure 2 continued on next page*

*Figure 2 continued*

saline (PBS) group. (**D**) The quantified analysis of migration displacement, distance, and velocity of tracked CTLs vs. time (min) based on panel (**C**). (**E**) Proportion of high-crosslink Col IV in lung tissue of mice exposed to FPM or PBS for 7 days, which was calculated by the 'high-cross' Col IV fragment divided by the sum of different fractions ('low-cross' ones and 'high-cross' ones). The content of each part was detected by ELISA. n = 5. (**F**) Western blotting analysis of 'low-cross' collagen and 'high-cross' collagen in lung tissue of mice exposed to FPM or PBS for 7 days. (**G–J**) The in-depth analysis of representative Col IV immunofluorescence images of lung tissue in the mice exposed to FPM for 7 days through ImageJ. (**G**) Look-up tables (LUTs) analysis of Col IV fluorescence intensity. (**H**) Surface plot analysis of Col IV distribution based on invert binary distance maps. (**I**) Binary images of Col IV network generated by ridge detection plugin. (**J**) Quantification analysis of junction number and density in Col IV network based on panel (**I**). n = 3. (**K**) Representative immunofluorescence images of Col IV and SEM images of FMP-exposed lung tissue treated with collagenase D (50 μg/mL). Scale bar = 10 μm. (**L**) The trajectory of CTLs migrating in FPM-exposed lung tissue slice treated as for panel (**K**). (**M**) Average migration displacement, distance, and velocity of tracked CTLs vs. time (min) in lung tissue slice treated as in panel (**K**). Images are representative of three independent experiments. Results are shown as mean ± SD. *p<0.05 after ANOVA with Dunnett's tests.

The online version of this article includes the following video, source data, and figure supplement(s) for figure 2:

**Figure supplement 1.** Transcriptional level of typical markers related to cytotoxic T lymphocytes' (CTLs') migration, integrin-1 (ITGB-1), C-X-C motif chemokine receptor 3 (CXCR 3), and Rho-associated kinase (ROCKi) in Jurkat T cells after they were stimulated with fine particulate matter (FPM) for 48 hr (**A**) or in the CTLs separated from lung tissue exposed to FPM for 7 days (**B**).

**Figure supplement 2.** The quantified analysis of pore diameter of interstitial matrix in the lung tissue, based on the scanning electron microscope (SEM) images and analyzed with ImageJ.

**Figure supplement 3.** Schematic diagram of analyzing of cytotoxic T lymphocytes' (CTLs') migration in lung tissue slice of fine particulate matter (FPM)-exposed mice or phosphate buffer saline (PBS) group.

**Figure supplement 4.** Schematic diagram of separating collagen fraction with different crosslink level.

**Figure supplement 5.** The relative intensity of high-crosslink Col IV to low-crosslink ones according to the Western blotting (WB) results of lung tissue exposed to fine particulate matter (FPM) for 7 days based on the ImageJ analysis.

**Figure supplement 6.** Representative Col IV immunofluorescence images of lung tissue in the mice exposed to fine particulate matter (FPM) for 7 days, with the blue DAPI staining images shown in the inserted box.

**Figure supplement 7.** Analysis of crosslinking level of Col I and Col III in lung tissue of mice exposed to FPM.

**Figure supplement 8.** The correlation analysis of CTLs' migration index with the level of Col IV crosslink.

**Figure supplement 9.** Schematic diagram about cytotoxic T lymphocytes' (CTLs') migration in the lung tissue exposed to fine particulate matter (FPM).

**Figure 2—video 1.** Dynamic migration video of T cells in the slice of native lung tissue.
https://elifesciences.org/articles/75345/figures#fig2video1

**Figure 2—video 2.** Dynamic migration video of T cells in the slice of lung tissue exposed to PM1.
https://elifesciences.org/articles/75345/figures#fig2video2

**Figure 2—video 3.** Dynamic migration video of T cells in the slice of lung tissue exposed to TSP.
https://elifesciences.org/articles/75345/figures#fig2video3

**Figure 2—video 4.** Dynamic migration video of T cells on PM1-exposed lung tissue pretreated with with PBS.
https://elifesciences.org/articles/75345/figures#fig2video4

**Figure 2—video 5.** Dynamic migration video of T cells on PM1-exposed lung tissue pre-treated with collagenase D.
https://elifesciences.org/articles/75345/figures#fig2video5

**Figure 2—video 6.** Dynamic migration video of T cells on TSP-exposed lung tissue pre-treated with PBS.
https://elifesciences.org/articles/75345/figures#fig2video6

**Figure 2—video 7.** Dynamic migration video of T cells on TSP-exposed lung tissue pre-treated with collagenase D.
https://elifesciences.org/articles/75345/figures#fig2video7

**Source data 1.** Excel spreadsheet source file for *Figure 2D, E, J and M*.

*1986*. First, ELISA was performed after different fragments of collagens were respectively harvested and divided into two categories by a reported protocol (*Popov et al., 2011*), that is, 'low-crosslinked' ones (low-cross) – containing freshly secreted collagens, procollagens, and moderately crosslinked collagens – and the other remainder 'high-crosslinked' ones (high-cross) (*Figure 2—figure supplement 4*). The results showed both PM1 and TSP exposure significantly elevated the high-crosslink proportion of Col IV, twofold higher than that in the PBS group, based on the separate examination of low-cross and high-cross ones (*Figure 2E*). Next, the relative quantification of high-crosslinked collagens compared with low-crosslinked ones based on Western blotting (WB) analysis showed consistent changes (*Figure 2F*, *Figure 2—figure supplement 5*). Besides, the IF images indicated that FPM

exposure induced Col IV in the lung tissue to generate enhanced crosslink and denser distribution, leading to the collagen network with more junction site and higher junction density (*Figure 2G–J*, *Figure 2—figure supplement 6*). However, the other two types of collagens (Col I and Col III) showed no obvious change (*Figure 2—figure supplement 7*), demonstrating an increased Col IV crosslinking accounted for the change in the lung ECM structure. Furthermore, based on the related integrated optical density (IOD) of Col IV in lung tissue slices and related CTLs' migration index (migration distance, displacement, and velocity) of different groups in *Figure 2D*, we performed the Pearson's correlation analysis and found an inverse relationship between Col IV density and CTLs' migration potential (*Figure 2—figure supplement 8*). Furthermore, we pretreated the FPM-exposed lung tissue with collagenase D to reduce the Col IV crosslink and alleviate Col IV density (*Figure 2K*). The trajectory images and related quantification analysis showed that CTLs' migration was effectively recovered (*Figure 2L and M*, *Figure 2—videos 4–7*), further validating the crucial role of Col IV crosslink on the CTLs' movement. These data together suggested that FPM exposure blocked CTLs migration and trapped these cells mainly through increasing Col IV crosslinking and consequently generating a denser ECM in the lung tissue, which might isolate the tumor cells from the CTLs' attack (*Figure 2—figure supplement 9*).

## FPM increases Col IV crosslinking through promoting sulfilimine bond formation

We then investigated why FPM exposure led to increased Col IV crosslinking (*Gaggar and Weathington, 2016*). According to recent discovery, protein adsorbed onto the nanoparticles surface would endow them with new activities (*Monopoli et al., 2012*; *Wang et al., 2017*). As the median size of both PM1 and TSP is about 100–200 nm, it is possible that the collagen-crosslinking activity of FPM is derived from the proteins adsorbed onto their surface from lung tissue. To elucidate this, we separately incubated FPM in PBS or lung homogenate (LH) to simulate the scenario of FPM per se (FPM group) or the complex of FPM and its surface proteins (LH-FPM group, including LH-PM1 and LH-TSP). Then, according to an established experimental model with a slight modification (*Bhave et al., 2012*), soluble Col IV was generated by stimulating mouse bone marrow fibroblasts M2-10B4 cells, which highly express Col IV, with the inhibitor of collagen crosslink (*Figure 3—figure supplement 1*). Next, the effect of FPM on the crosslink of soluble Col IV in the cellular system and acellular system was respectively analyzed as shown in *Figure 3A*. For the cellular system, Col IV immunostaining result of M2-10B4 cells showed FPM itself could not induce the crosslinking of soluble Col IV (*Figure 3B*). Relatively, the LH-FPM initiated the crosslinking and reinforced the network to a greater extent than that induced by LH per se, which could be validated by intensive crosslink intensity and a denser Col IV distribution, and collagen network with more junction site and higher junction density (*Figure 3C–F*). Meanwhile, for the acellular experiment, the cell lysate of M2-10B4-enriched soluble Col IV was incubated with FPM or LH-FPM mixture. WB result showed a similar scenario – the naked FPM had little devotion to Col IV crosslink, but the LH-FPM dramatically enhanced the high-crosslinked Col IV fragment (*Figure 3G*, *Figure 3—figure supplement 2*).

These data raised the question of how LH-FPM increased Col IV crosslinking. During crosslinking, the triple-helical protomer of Col IV, as the building block, forms network through two key types of crosslinking sites (*Brown et al., 2017*): NC1 domains including sulfilimine bond (-S=N-) formed at the C-terminal (*Vanacore et al., 2009*) and 7S tetramers, including aldehyde group formed at the N-terminal (*Risteli et al., 1980*; *Figure 3H*). To distinguish which one is mainly disturbed by the FPM, we detected their changes under FPM stimulation respectively: (1) for the NC1 domain, sulfilimine bond (-S=N-) is formed by two juxtaposed Col IV protomers at residues methionine 93 (Met93) and hydroxylysine 211 (Hyl211) (*Figure 3—figure supplement 3*; *Bhave et al., 2012*). Based on indicated theoretical mass of crosslinked tryptic peptides containing -S=N in NC1 domain, we performed high-resolution liquid chromatography-mass spectrometry (LC-MS) analysis to differentiate these peptides. For the NC1 domain separated from the crosslinked Col IV as illustrated in *Figure 3H*, we found significantly more sulfilimine-containing peptides in LH-FPM-treated soluble Col IV than that in the LH group according to the total ion chromatography (TIC) diagram (*Figure 3I*). (2) For the 7S domain crosslinking site, it was derived from the oxidation of one lysine residue in the N-terminal to the aldehyde (*Anazco et al., 2016*). The generated allysine would subsequently undergo a series of condensation reactions with other amino acids, mainly the other lysine or lysines on neighboring C-terminus,

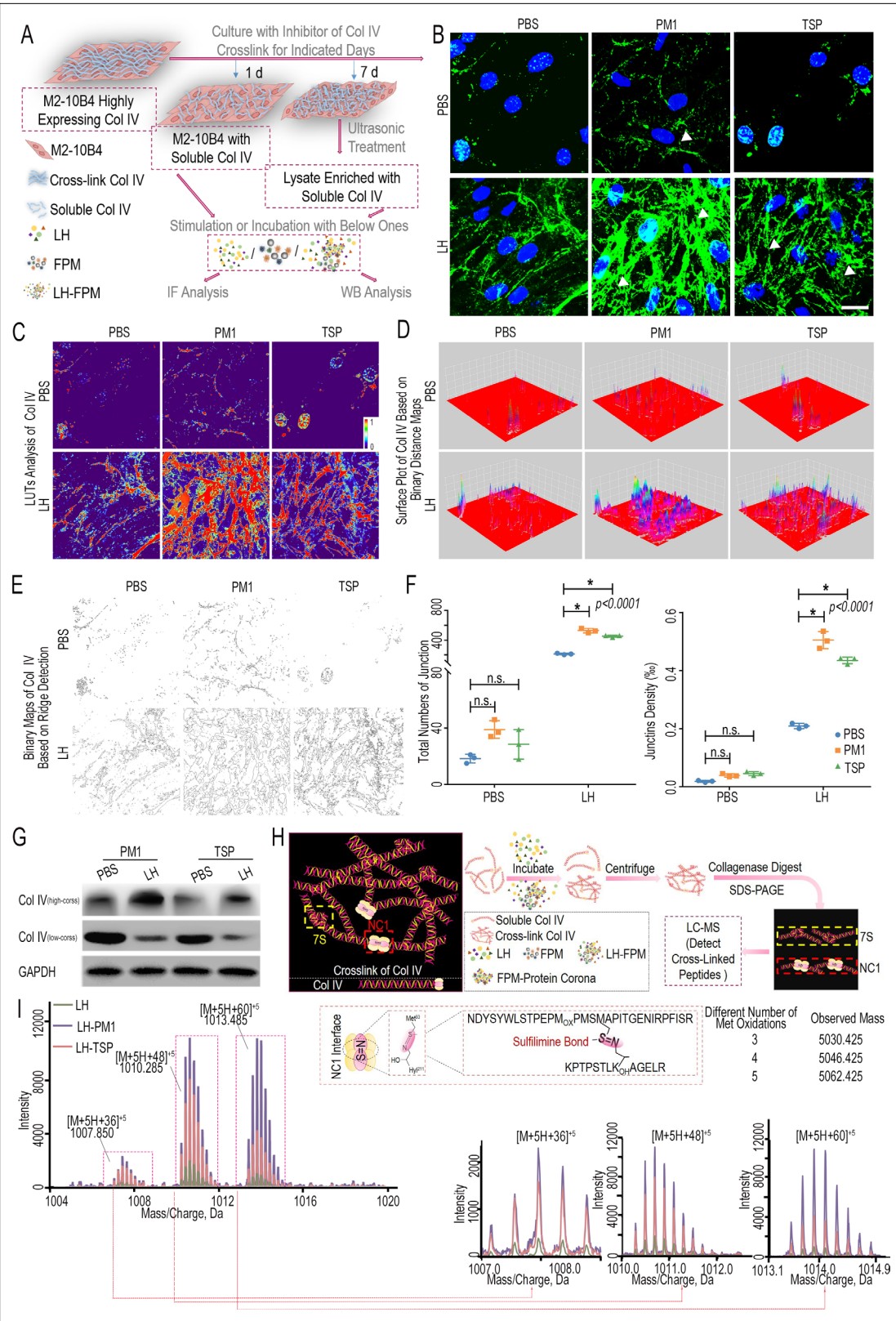

**Figure 3.** Fine particulate matter (FPM) increased Col IV crosslink via promoting sulfilimine bond formation at the NC1 domain. (**A**) Schematic representation of the procedures to generate soluble Col IV and analyze the effect of FPM on its crosslink. Briefly, the M2-10B4 cells highly expressing Col IV were treated with crosslink inhibitor for 1 or 7 days and then treated with FPM or the mixture of lung homogenate (LH)-FPM, stimulating the scenario of FPM per se or its interface with LH, to initiate the crosslink. The crosslink level of Col IV was analyzed with diversity methods. (**B**)

*Figure 3 continued on next page*

*Figure 3 continued*

Representative immunofluorescence capture of Col IV in M2-10B4 cells stimulated with FPM or LH-FPM for 24 hr after pretreated with crosslink inhibitor for 1 day. Scale bar = 20 μm. Images are representative of three independent experiments. (**C–F**) The in-depth analysis of Col IV immunofluorescence images in panel (**B**) through ImageJ. (**C**) Look-up tables (LUTs) analysis of Col IV fluorescence intensity. (**D**) Surface plot analysis of Col IV distribution based on invert binary distance maps. (**E**) Binary images of Col IV network generated by ridge detection plugin. (**F**) Quantification analysis of junction number and density in Col IV network based on panel (**E**). (**G**) Western blotting of 'low-cross' and 'high-cross' Col IV fraction in M2-10B4 cells lysate enriched with soluble collagen after their treatment with FPM or LH-FPM. (**H**) Schematic diagram of separating fragments containing the NC1 domain crosslink site in Col IV. The general crosslink network generated by Col IV is displayed on the left, with the important crosslink sites (7S domain and NC1 domain) respectively labeled in the yellow and red dotted box. (**I**) High-resolution mass spectrum depicting tryptic peptides containing sulfilimine bond (-S=N-), with magnified spectrum displayed on the bottom. The formation of -S=N- and the known peptide sequence with different oxidation containing the sulfilimine bond are shown on the upper right.

The online version of this article includes the following source data and figure supplement(s) for figure 3:

**Figure supplement 1.** Western blotting analysis of 'soluble' and 'crosslinked' Col IV fraction in M2-10B4 cells lysate after the cells were treated with different concentrations of crosslink inhibitor phloroglucinol (PHG) for 7 days.

**Figure supplement 2.** The relative intensity of high-crosslink Col IV to low-crosslink ones according to the Western blotting (WB) results in M2-10B4 cells lysate enriched with soluble collagen after their treatment with fine particulate matter (FPM) or the mixture of lung homogenate (LH) and FPM, that is, LH-FPM.

**Figure supplement 3.** Structure of sulfilimine bond formed at the covalent crosslinks of NC1 domains, shown in the lilac box.

**Figure supplement 4.** The analysis of allysine to reflect the 7S domain generated during the Col IV crosslinking.

**Source data 1.** Excel spreadsheet source file for *Figure 3F and I*.

forming methylenimine bond (-C=N-), pyridine, or others to stabilize crosslink (*Figure 3—figure supplement 4A*). During the process, the detection of primary product allysine could reflect the level of 7S domain crosslink. With the reported specific and efficient probes to allysine (*Waghorn et al., 2017*), the allysine yielded during the crosslinking of the soluble Col IV incubated with LH or LH-FPM was respectively analyzed. The result showed a slight difference, indicating the 7S domain would not be interfered by FPM (*Figure 3—figure supplement 4B*). Summarily, FPM gained a catalyzing activity from the proteins adsorbed from the tissue, which mediated the crosslinking by forming excessive -S=N- bonds among Col IV molecules.

## Phase transition of PXDN on FPM surface increases its activity for Col IV crosslinking

Although the above findings demonstrated that LH-FPM increased sulfilimine bond formation to enhance Col IV crosslinking, it remained unclear how FPM gained the activity from LH to mediate this biochemical process in vivo. To elaborately dissect this process, following a standard procedure (*Figure 4—figure supplement 1*; *Carrillo-Carrion et al., 2017*), we separated the biomacromolecules from LH-FPM, the majority of which are proteins and also known as 'protein corona' (*Monopoli et al., 2012*). Unexpected corona formation can trigger serious pathological reactions (*Wang et al., 2017*; *Tenzer et al., 2013*). Thus, we speculated whether FPM could recruit certain proteins related to Col IV crosslink into its corona, thereby enriching and empowering this protein – to influence the crosslinking of collagen IV.

Given that the sulfilimine bond is uniquely catalyzed by PXDN enzyme in animal tissue (*Bhave et al., 2012*; *Weiss, 2012*), we focused and detected the PXDN in FPM's protein corona. The LC-MS result showed that PXDN was listed in the component profile of protein corona on both PM1 and TSP (*Figure 4—source data 1*), reflecting the interaction of PXDN and FPM. We further analyzed PXDN adsorbed on FPM and its time evolution with WB (*Wang et al., 2017*). The data showed that PXDN was not only adsorbed on the FPM (*Figure 4A*), but also stably tethered to FPM as the incubation time increased (*Figure 4B*), underlying that it might affect FPM's biological behavior durably. To further assess the adsorption of FPM to PXDN, we injected rhodamine fluorescence-labeled particles (R-FPM) via the trachea into the lung tissue and detected PXDN therein. The IF co-localization of FPM and PXDN in vivo was clearly presented, confirming the recruitment of this enzyme to FPM (*Figure 4C*), which could be further validated by the dramatically similar distribution of PXDN and FPM on M2-10B4 cells (*Figure 4D*). Taken together, these data indicate that FPM enriches and stabilizes PXDN in its surface corona.

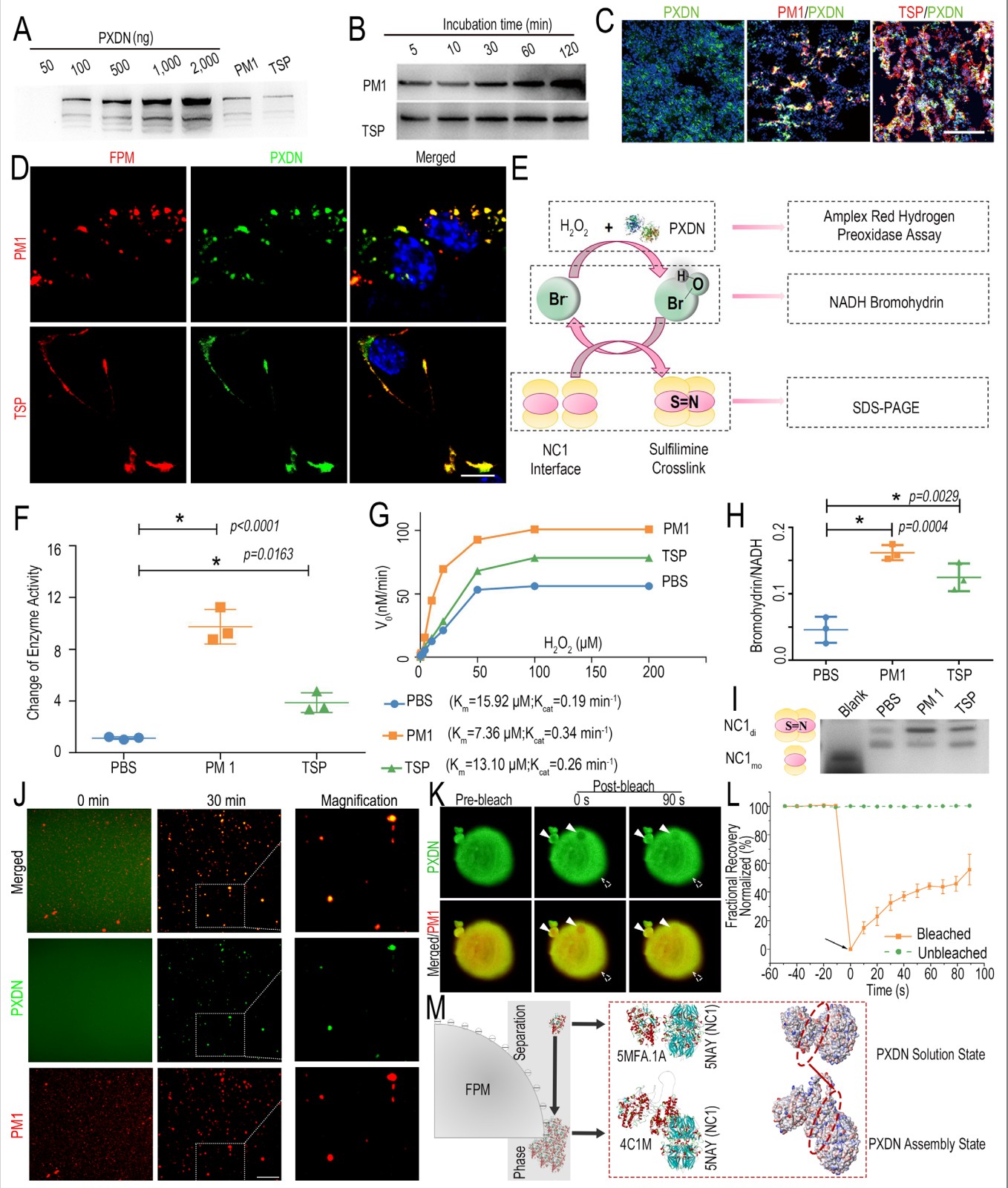

**Figure 4.** Fine particulate matter (FPM) increased peroxidasin (PXDN) activity by triggering the enzyme's phase transition. (**A**) Quantitative Western blotting analysis of PXDN harvested from FPM corona formed in lung homogenate (LH) after incubation for 2 hr, with a serial content of recombinant PXDN protein as the standard control. (**B**) Western blotting analysis of PXDN at indicated time points to identify its time evolution in FPM's protein corona. (**C**) Representative confocal microscopic photographs showing the co-localization of rhodamine-labeled FPM (shown in red) and PXDN in

*Figure 4 continued on next page*

*Figure 4 continued*

lung tissue. PXDN is indicated as green. Scale bar = 100 μm. (**D**) Representative fluorescent photographs of M2-10B4 cells treated with rhodamine-labeled FPM and fluorescein isothiocyanate (FITC)-labeled PXDN for 1 hr. Rhodamine-labeled FPM is shown in red, FITC-PXDN in green, and DAPI staining for the nuclei in blue. Scale bar = 10 μm. (**E**) Schematic representation of the procedure for the formation of sulfilimine bond catalyzed by PXDN. Aimed at the substrate $H_2O_2$, the intermediates HOBr and the final NC1 domain with sulfilimine bond, different experimental analyses were respectively performed. (**F, G**) Fold change of enzyme activity (**F**) and enzyme kinetics (**G**) of PXDN stimulated with FPM, determined with Amplex Red Hydrogen Peroxidase Assay Kit. (**H**) Ration of NADH bromohydrin relative to NADH based their intensity of peaks detected by liquid chromatography-mass spectrometry (LC-MS). The analysis was performed after PXDN was incubated with FPM for 30 min and then catalyzed in the presence of 100 μM $H_2O_2$ and 200 μM NaBr at 37°C for 30 min. (**I**) SDS-PAGE and Coomassie staining of NC1 domain 4 hr after they were incubated with PXDN, following the latter's incubation with PBS or FPM for 30 min. The crosslinked dimeric ($NC1_{di}$) and un-crosslinked monomeric subunits ($NC1_{mo}$) are respectively labeled. Images are representative of three independent experiments. (**J**) The confocal microscopy of FITC-labeled PXDN was incubated with rhodamine-labeled FPM in LH for the indicated time (0 and 30 min). Shown at the right are images with higher magnification for the assemblies of PXDN's liquid-like droplets on the FPM at 30 min. Scale bar = 5 μm. (**K**) Representative images from fluorescence recovery after photobleaching (FRAP) experiments showing the dynamic and reversible characteristics of PXDN droplets. The rhodamine-labeled FPM is shown in red and FITC-labeled PXDN in green. The bleached region of interest (ROI) is indicated with white triangles, and the unbleached control ROI is labeled with dotted white ones. (**L**) Quantification of fluorescence recovery percentage in the ROIs of PXDN's liquid-like droplets. The black arrow indicates the initiation of laser bleach treatment. (**M**) Interactive docking model on the effect of PXDN's phase separation on its enzymatic performance at the catalytic interface of NC1 domain. The structures of PXDN solution state (PDB ID: 5MFA.1) and its assembly (PDB ID: 4C1M; created through homology modeling) are respectively displayed as the lateral stereo view of transparent chain model (left) and SWISS-MODEL (right). The interface site of contact between NC1 domain (PDB ID: 5NAY) and PXDN is labeled with the red dotted ellipses. n = 3. Results are shown as mean ± SD. *$p<0.05$ after ANOVA with Dunnett's tests.

The online version of this article includes the following source data and figure supplement(s) for figure 4:

**Figure supplement 1.** Schematic diagram of separation and preparation of fine particulate matter's (FPM's) protein corona in lung homogenate (LH).

**Figure supplement 2.** Liquid chromatography-mass spectrometry (LC-MS) spectrum for NADH (dotted line) and the bromohydrin (line), according to the reported literature (*Bathish et al., 2018*).

**Figure supplement 3.** HOCl production induced by peroxidasin (PXDN) measured with HOCl detecting fluorescent probes after the enzyme was incubated with fine particulate matter (FPM) for 30 min and then catalyzed in the presence of 100 μM $H_2O_2$ and 200 mM NaCl, with a serial concentration of HOCl as internal control.

**Figure supplement 4.** Effect of fine particulate matter (FPM) on peroxidasin's (PXDN's) phase separation.

**Figure supplement 5.** Intrinsically disordered region (IDR) analysis of peroxidasin (PXDN) domains by the IUPred algorithm.

**Source data 1.** List of protein components identified by liquid chromatography-mass spectrometry (LC-MS) for PM 1's and tobacco smoke particulates' protein corona.

**Source data 2.** Excel spreadsheet source file for *Figure 4F–H,L*.

The adsorption of FPM might disturb the activity of proteins, especially for the enzyme therein. Thus, we examined the activity of FPM-recruited PXDN as illustrated in *Figure 4E*. Aimed at the substrate $H_2O_2$, the intermediates hypohalous acids, and the final NC1 domain with sulfilimine bond, different experimental analyses were respectively performed (*Bhave et al., 2012*). First, peroxidase activity of PXDN incubated with FPM was measured through Amplex Red molecular probes (*Bathish et al., 2018*). The relative fluorescence intensity showed that FPM incubation raised the enzymatic activity of PXDN up to 5- to 10-fold (*Figure 4F*). Next, PXDN's enzyme kinetic behaviors were investigated according to the Michaelis–Menten model. Based on the generated Line weaver–Burk representative plot, the Michaelis constant (Km) and turnover number (Kcat) were calculated, which respectively reflects the enzyme-substrate binding efficiency and the enzymatic efficiency. The result showed that PXDN incubated with FPM revealed significantly decreased Km (PM1: 7.36 μM; TSP: 13.10 μM vs. PBS: 15.92 μM) and increased catalytic efficiency about 1.4- to 1.8-fold than the enzyme per se (*Figure 4G*). Also, the secondary product hypohalous acids (*Bhave et al., 2012*; *McCall et al., 2014*), for instance, HOBr and HOCl, generated by PXDN from bromide and chloride, were respectively analyzed. For HOBr, bromohydrin formed by the bromination of NADH was measured based on its close relation with HOBr production as reported in the literature (*Bathish et al., 2018*; *Soudi et al., 2015*). The LC-MS detection showed that the ratio of bromohydrin to NADH in the group of PXDN incubated with FPM was three- to fivefold higher than that of the enzyme per se group, indicating more HOBr production (*Figure 4H*, *Figure 4—figure supplement 2*). Besides, we measured HOCl in consideration of the vast excess of Cl⁻ over Br⁻ in most animals (*Weiss et al., 1986*), although PXDN uses Br⁻ to catalyze the formation of sulfilimine crosslinks with greater efficiency (*McCall et al., 2014*). With the detection of a hypohalous acid-detecting fluorescent probe (*Xing et al., 2018*), we found

that HOCl produced by the FPM-incubated PXDN was about two- to fourfold higher than that of the PBS group (*Figure 4—figure supplement 3*). Furthermore, to analyze the effect of FPM's adsorption on the catalytic product, after the commercial non-crosslinking NC1 domain (NC1$_{mo}$) was reacted with FPM-incubated PXDN, the generated crosslinked dimeric with sulfilimine bond (NC1$_{di}$) therein was detected by sodium dodecyl sulfate polyacrylamide gel electrophoresis (SDS-PAGE) (*Bhave et al., 2012*; *Fidler et al., 2014*). The result suggested that FPM incubation significantly enhanced PXDN's enzymatic performance, with a higher yield of crosslinked NC1 dimeric subunits (*Figure 4I*). Taken together, these results suggest that the pro-crosslink potential of FPM is attributed to the aberrant enzyme activity of PXDN adsorbed on its surface.

Next, we gave an insight into the mechanism of PXDN's tampered catalysis. Recent emerging evidence suggests that phase transition (or separation) is a common way to regulate proteins' activity (*Huang et al., 2019*). Phase transition refers to the process that macromolecular solution condenses into liquid droplets, solids, or gels (*O'Flynn and Mittag, 2021*; *Boeynaems et al., 2018*) in response to certain physicochemical stimuli, sharply increasing the macromolecule's concentration and separating them from the surrounding compartments (*Alberti et al., 2019*; *Peeples and Rosen, 2021*). Such enrichment affects the subsequent biochemical reactions. Thus, we asked whether the FPM's adsorption could induce the PXDN's phase transition, thus disturbing the latter's enzymatic activity. Taking PM1 as an example, with a series of microscopic observations, we found the formation of PXDN's liquid-like droplets in LH after its incubation with PM1 with both confocal fluorescence and phase-contrast microscope (*Figure 4J*, *Figure 4—figure supplement 4A*). However, for the PXDN alone in LH under the same processing time and imaging parameters, no assemblies are observed. Moreover, the profiles of fluorescence distribution further indicated the phase-separating PXDN's accumulation was initialized on the PM1, based on their evident colocalization (*Figure 4—figure supplement 4B*). Besides, to characterize the dynamic nature of the droplets, we performed fluorescence recovery after photobleaching (FRAP) experiments. FRAP studies revealed that fluorescence of the bleached region of PXDN droplets could be partially recovered in minutes after photobleaching (*Figure 4K and L*). The reversible characteristic observed for PXDN droplets on PM1 further validated that PXDN underwent phase separation. More interestingly, the droplet-like accumulation of PXDN on FPM could also be observed in M2-10B4 cells (*Figure 4D*). Overall, these results indicated that the increased activity for PXDN's crosslinking Col IV was triggered by its phase transition on the FPM surface.

Furthermore, we theoretically inferred the relationship of PXDN's phase separation with its enzymatic performance. First, we focused on the low-complexity domains in PXDN sequence, which could drive phase transition and be predicted by intrinsically disordered regions (IDRs; *Figure 4—figure supplement 5*). The analysis indicated that PXDN's phase separation might occur at sequence 200–400 aa, which shows a higher IDR score (*Alberti et al., 2019*). More importantly, it is away from PXDN's activity center (800–1200 aa) according to the spatial structure in SWISS-MODEL, giving us a hint that PXDN's phase transition on FPM surface might not lead to deformation of PXDN's catalytic center and loss of its function. Next, protein-protein interactive docking simulation was performed at the interface of PXDN with its substrate NC1 domain through ZDock protocol (*Figure 4M*). As a homotridmeric multidomain enzyme, PXDN exists as trimerization in solution, and three monomers of PXDN are linked by disulfide bonds at the indicated flexible linker region in the residual noncatalytic domains (*Bathish et al., 2018*). According to the reported modeled structure of PXDN (*Paumann-Page et al., 2020*), its trimeric peroxidase domain displayed a triangular arrangement. However, oversized trimerization of PXDN might not get in contact thoroughly with its substrate (*Lázár et al., 2015*). So we chose the PXDN monomer with the exposed enzymatic surface contacting with NC1 domain to simulate its function at solution state. The computational result indicated that once PXDN triggered phase separation, which transferred from the solution state to the aggregation one (the putative assembly structure created through homology modeling) (*Bernardes et al., 2015*), the interactive area at the NC1 interface would be notably increased, thus facilitating the enzymatic catalysis.

## Inhibiting PXDN ameliorates FPM-induced tumorigenesis

Based on the above findings, we speculated that inhibiting PXDN could abolish ECM change and recover CTLs migration in the lung. To testify to our hypothesis, the effect of the plasmids capable of ectopically expressing PXDN-specific short hairpin RNA (PXDN shRNA, shPXDN) was detected.

After validating its interference efficiency (*Figure 5—figure supplement 1*), shPXDN mixed in the in vivo-jetPEI gene transfer regent was delivered into murine lung tissue through trachea injection, as illustrated in *Figure 5—figure supplement 2*. The assessment of Col IV with a different fraction ('low-cross' ones and 'high-cross' ones) measured by ELISA revealed that the shPXDN diminished crosslinked level of Col IV (*Figure 5A*). Masson's trichrome staining images and SEM observation also confirmed that the shPXDN effectively decreased collagen density and expanded interstitial space (*Figure 5—figure supplement 3*). More importantly, the ameliorative Col IV network induced by shPXDN further recovered the migration and accumulation of CTLs in lung tissue 1 day after the LLC stimulation, revealed as the CTLs' migration trajectory images and the immunofluorescent staining (*Figure 5B and C*). Flow cytometry analysis further demonstrated that shPXDN efficiently reversed the CTLs' infiltration in the FPM-exposed lung tissue (*Figure 5D*, *Figure 5—figure supplement 4*).

These results inspired us to assess the effect of shPXDN in suppressing the tumor progress induced by FPM, as illustrated in *Figure 5—figure supplement 5*. Encouragingly, our data showed that the shPXDN significantly suppressed tumor growth, diminished tumor grade, and reduced tumor number and burden (*Figure 5E and F*, *Figure 5—figure supplement 6*) in both $Kras^{G12D}Trp53^{-/-}$ transgenic model and LLC model. These results suggested the feasibility of inhibiting PXDN as a potential therapeutic target for FPM-associated lung cancer. Moreover, the efficacy of small-molecule PXDN inhibitors, including methimazole (MMZ) and phloroglucinol (PHG) (*Bhave et al., 2012*), was similarly studied with optimal therapeutic doses (*Figure 5—figure supplement 7*). To be satisfactory, both MMZ and PHG also effectively suppressed lung tumorigenesis (*Figure 5G–J*, *Figure 5—figure supplement 8*), further expanding the strategy for lung cancer treatment.

## Discussion

Although inhalable particles from air pollutants and tobacco smoke have clearly been identified as a potent carcinogen to humans, its pathological mechanism remains unclear, which hampers the design of therapeutic approaches. Existing findings, though supporting that FPM induces lung cancer, provide insufficient and inconsistent explanations for the underlying mechanism. In this study, we have discovered an unexpected mechanism that, as shown in *Figure 6*, apart from directly targeting tumor cells, the inhaled FPM changes the formation of lung tissue matrix to prevent the infiltration of T lymphocytes and their antitumor immunosurveillance, which consequently accelerates lung tumorigenesis.

Our study was inspired by an unanswered, fundamental question – which component in the lung tissue is first affected by FPM? Most previous studies suggest that FPM directly acts on cells by inducing gene mutation and increasing malignant 'seed cells' in the affected lung (*Somers et al., 2004*). They further suggest that the particles carry chemical reagents, such as acrolein, nicotine, oxidants, and reactive nitrogen moieties (*Hecht, 2002*), into the lung to exert such effects. However, the respiratory tract is exposed to various environmental substances (e.g., $O_2$ or $O_3$) that are much more powerful in damaging DNA, more abundant in the air, and more soluble in the tissue than the above chemicals. It requires further explanation of how these assumed mutagens, delivered in a trace amount by the particles into the lung tissue, could significantly increase lung cancer incidence. Moreover, FPM from different locations carries distinct types of those carcinogens existing in the local environment, but these highly various particles exert the same carcinogenic effect (*Kelly and Fussell, 2012*), which is unexplainable. Additionally, in vitro, FPM is considered more capable of entering the cell than larger particles (*Gratton et al., 2008*) (such as the PM10, i.e., the particulate matter with the diameter of 1–10 μm); but in vivo, particles are difficult to directly contact with cells due to the barrier effects from sticky body fluids and gel-like ECM proteins (*Fernandes et al., 2009*). Since inhaled particles are mostly accumulated in the ECM (*Engin et al., 2017*), there is no reason to ignore the impact of ECM and only focus on the interaction between particles and cells. Thus, we suspected that the lung tissue ECM, apart from the cells, could also be affected by the particles. Indeed, ECM abnormality is closely correlated with cancer development. On the one hand, the clinical samples showed the developed tumor was surrounded by a dense collagen matrix; on the other hand, one typical example is that tissue fibrosis often precipitates tumor development in the same tissue – such as liver cirrhosis progressing into hepatoma and pulmonary fibrosis leading to lung cancer (*Cox and Erler, 2016*; *Tzouvelekis et al., 2020*). These clinical and experimental proven reports prompt us to investigate the pathological change of FPM-stimulated lung interstitial ECM and its role in lung tumorigenesis.

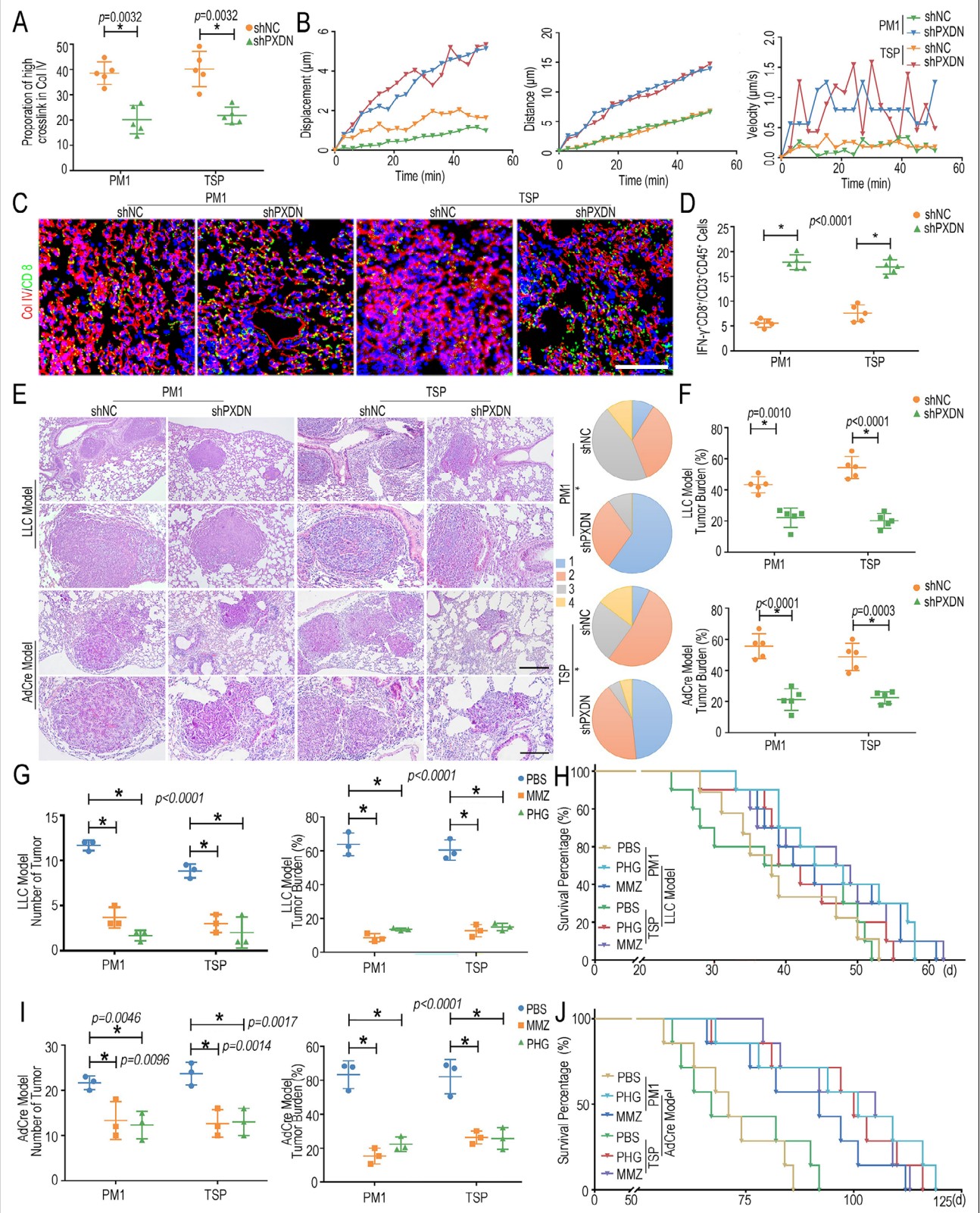

**Figure 5.** Peroxidasin (PXDN) inhibitor alleviated fine particulate matter (FPM)-induced lung tumorigenesis. (**A**) Proportion of high-crosslink Col IV in lung tissue of FPM-exposed mice pretreated with PXDN-specific short hairpin RNA (PXDN shRNA, shPXDN), which was calculated by the 'high-cross' Col IV fragment divided by the sum of different fractions ('low-cross' ones and 'high-cross' ones) based on ELISA. (**B**) Migration displacement, distance, and velocity vs. time (min) of tracked cytotoxic T lymphocytes (CTLs) in lung tissue slice of FPM-exposed mice administrated with PXDN

*Figure 5 continued on next page*

*Figure 5 continued*

shRNA (shPXDN). (**C**) Representative immunofluorescence images of CTLs' infiltration into the FPM-exposed lung tissue pretreated with shPXDN 1 day after intravenous injection of Lewis lung carcinoma (LLC). Scale bar = 100 μm. (**D**) The statistical analysis of CTLs (IFN-γ⁺CD8⁺/CD45⁺CD3⁺) based on flow cytometry in lung tissue of mice treated as in panel (**C**). n = 5. (**E**) Representative hematoxylin and eosin (H&E) staining images of lung tissue (the lower ones: captured with higher magnification) yielded from LLC model and *Kras^G12D^Trp53^-/-* model after mice pretreated with shPXDN. Scale bar = 200 μm (upper) and 100 μm (lower). Tumor stage (stages 1–4) in lungs of *Kras^G12D^Trp53^-/-* mice is shown on the right. p-Values are for comparisons of the percentage of stage 3 and 4 tumors in different groups. n = 5. (**F**) Statistical analysis of tumor burden of mice in LLC model and *Kras^G12D^Trp53^-/-* model administrated with shPXDN. n = 5. (**G–J**) Statistical analysis of number and burden of tumors and survival curve of mice in LLC model (**G, H**) and *Kras^G12D^Trp53^-/-* model (**I, J**) administrated with methimazole (MMZ) or phloroglucinol (PHG). Images are representative of three independent experiments. n = 3. Results are shown as mean ± SD. *$p < 0.05$ after ANOVA with Dunnett's tests.

The online version of this article includes the following source data and figure supplement(s) for figure 5:

**Figure supplement 1.** The interference efficiency of PXDN-specific short hairpin RNA detected in cellular and lung tissue experiment.

**Figure supplement 2.** Schematic diagram of analyzing the effect of peroxidasin (PXDN)-specific short hairpin RNA interference (PXDN shRNA, shPXDN) on the structure of fine particulate matter (FPM)-exposed lung tissue.

**Figure supplement 3.** The histological analysis and SEM images of FPM-exposed lung tissue administrated with shPXDN.

**Figure supplement 4.** Representative flow cytometry analysis of cytotoxic T lymphocytes' (CTLs') infiltration (IFN-γ⁺CD8⁺/CD45⁺CD3⁺) into lung tissue of fine particulate matter (FPM)-exposed group pretreated with short hairpin peroxidasin (shPXDN) 1 day after they were stimulated with the Lewis lung carcinoma (LLC).

**Figure supplement 5.** Schematic diagram of administration of shPXDN on lung tumor related to FPM exposure.

**Figure supplement 6.** Therapeutic effect of shPXDN on lung tumor related to FPM exposure.

**Figure supplement 7.** Representative hematoxylin and eosin (H&E) staining images of lung tissue with intratracheal injection with different concentrations of methimazole (MMZ) and phloroglucinol (PHG) twice every three days.

**Figure supplement 8.** Administration and therapeutic effect of PXDN inhibitors on lung tumor related to FPM exposure.

**Source data 1.** Excel spreadsheet source file for *Figure 5*.

Our experimental findings validated this assumption by demonstrating – and elucidating the mechanism of – FPM-induced alteration in the interstitial lung structure. As soon as we, for the first time, found that FPM in the LH dramatically enhanced the crosslinking of soluble Col IV, we speculated that a specific biomolecule in LH enriched on the FPM surface mediates this action. With proteomics tools, we identified PXDN; this enzyme, specialized in catalyzing collagen network formation (*Bhave et al., 2012*), is enriched in the protein corona of FPM, enabling FPM to mediate ECM remodeling. Meanwhile, for PXDN, its adsorption onto FPM triggered its aberrant enzymatic activity through a 'phase-transition' process, which further disturbs Col IV crosslinking and the organization of the lung interstitial space. Pretreatment using PXDN inhibitor could alleviate the tumor-promoting effect of FPM. Therefore, an aberrant ECM remodeling mediated by enriched PXDN in the particle corona is the key pathological change caused by FPM.

Our study further elucidates how such a change in ECM accelerates tumor development. As a consequence of FPM treatment, the denser Col IV and more compacted interstitial space in lung tissue would hinder the migration of CTLs, which are the most 'informed' defender to protect against potential cancer and delay malignant progression (*DuPage et al., 2011*; *Krummel et al., 2016*). Once CTLs' migration track was blocked, the CTLs' infiltration would be impaired, thus decreasing the chance of cell-cell cytotoxic interaction to tumor cells in lung tissue (*Hanahan and Coussens, 2012*). According to the correlation between tumor mutation burden and immune infiltration in lung tumors (*Zhang et al., 2022*; *Thorsson et al., 2018*; *Helleday et al., 2014*), we speculated that the deficiency further aggravated the mutation of cancerous cells in the lung tissue. The insufficiency of CTLs' early response to tumor cells results in weakened immune surveillance and unmonitored mutation that in turn accelerate tumorigenesis. All the evidence recapitulates how FPM accelerated the development of lung cancer by interfering with the lung immune microenvironment, shedding light on the sequential consequences between particle-induced ECM remodeling, impaired immunosurveillance, and tumorigenesis.

Taken together, our study reveals a completely new mechanism by which inhaled fine particles promote lung tumor development. This mechanism is notable in three aspects. First, we herein provide direct evidence that protein corona on those foreign objects can elicit such a significant and

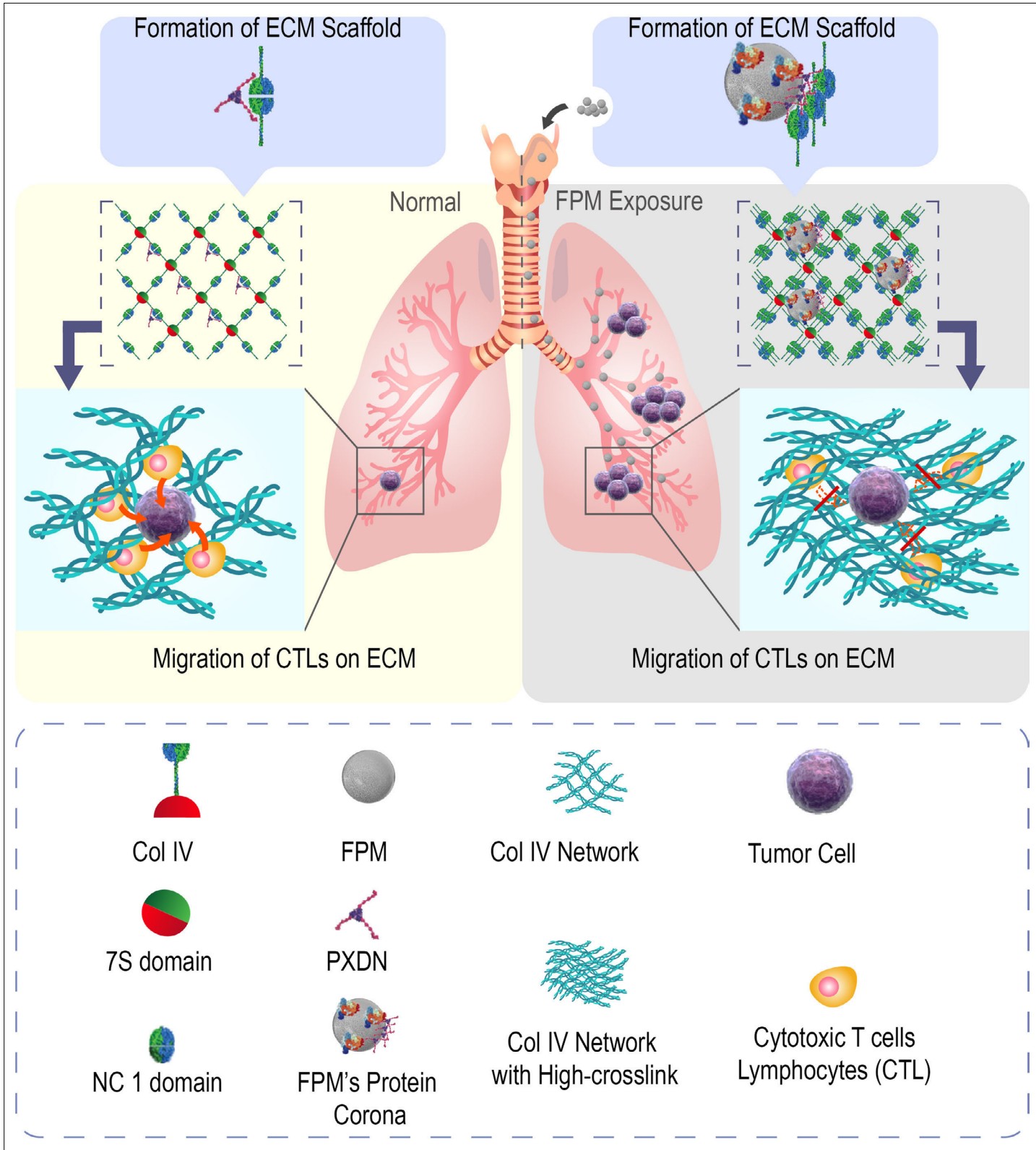

**Figure 6.** Schematic diagram of the mechanism underlying how fine particulate matter (FPM) promotes lung tumorigenesis. The adsorption of FPM triggers the phase separation of peroxidasin (PXDN) to generate an aberrant catalytic activity and induce a disordered crosslink of Col IV. Reinforced Col IV network impaired cytotoxic T lymphocytes' (CTLs') migration and local immunosurveillance, which considerably increases tumorigenesis in lung tissue.

adverse effect. Although corona formation on nanoparticles has been extensively studied in recent years, its involvement in a pathogenic process related to a global health issue is rare. Our findings here highlight the importance of the corona-endowed, 'new' enzymatic bioactivity of nanomaterials in vivo – and in a particular tissue. The data suggests that coronas adsorbed from the environment could also be catalytic, other than simply transforming the cellular and higher-level interactions, in agreement with other literature (*Dawson and Yan, 2021*; *Shang et al., 2021*). Second, PXDN, or other proteins mainly engaged in ECM modulation, is less expected as a major player in lung carcinogenesis, especially at the initial stage, until this study reveals it as a specific and unexpected molecular target for FPM. These investigations enable the specific design of PXDN-targeted preventive or therapeutic approaches. The relationship between the physicochemical properties of FPM and the affected PXDN activity should be further explored in greater detail. Third, in a specific organ (the lung), our data demonstrate that physical blockage of immunocytes movement directly increases tumorigenesis, which suggests an important previously unconsidered role of the in vivo delivering or deploying the power of the immune system in various immune-oncology processes.

## Materials and methods
### FPM collection and preparation
FPM samples within sizes of 1 μm were collected using TH-16A multiple-channel atmospheric particulate automatic sampler (Wuhan Tianhong Instrument Ltd., Wuhan, China) and filtered through Whatman PTFE membranes (GE Healthcare Life Sciences, Pittsburgh, USA). For particles in air pollutants (PM1), samples were conducted continuously for 7 days at different representatives of Nanjing City (Qixia, Jiangning, Pukou, Gulou, and Gaochun) in Jiangsu Province, Suzhou City in Anhui Province, and Tieling City in Liaoning Province (named as QX, JN, PK, GL, GC, SZ, and TL). For TSP samples, the residual smoke of burned tobaccos with filters was collected in a customized confined space. Then, PTFE filter membranes containing FPM were cut into 0.1 cm × 0.1 cm pieces, immersed in distilled water for 2 days, and oscillated ultrasonically for 1 hr three times. Detached FPM was separated with filter membranes after centrifugation at 2000 rpm for 5 min three times. Supernatant-enriched FPMs were vacuum freeze-dried and then stored at –20°C. For the preparation of FPM suspension, FPM samples were suspended in sterile 1× PBS (phosphate buffered saline) to achieve 10 mg/mL particles for further analysis.

### Characterization of FPM
A series of tests were performed to thoroughly characterize the nanoparticles. First, to analyze morphology of particles, dried PTFE filter membranes containing FPM were conducted with scanning electronic microscope (SEM) microscope LEO1530VP (JEOL Ltd., Tokyo, Japan). Second, all the particles were characterized for their Zeta potential and particle size using NanoSight NS300 instrument (Malvern Instruments, Malvern, UK). Third, to obtain essential information on the nanoparticles' size and shape, transmission electron microscopy (TEM) was carried out. After a few drops of deionized water-dispersed nanoparticles were dropped on the 300-mesh carbon-coated copper grid, TEM images of each sample were collected using TEM (JEOL Ltd.). Besides, the elemental analysis was performed on an element analysis instrument (Vario Micro Cube, Elementar, Germany), with the top 15 listed.

### Cell preparation and culture
Cell lines' culture

Lewis lung cancer cell lines (LLC), mouse bone marrow fibroblasts M2-10B4, and Jurkat T cells were obtained by Stem Cell Bank, Chinese Academy of Sciences, Shanghai, China. LLC cells expressing OVA peptide residues 257–264 (OVA$_{257-264}$) in the context of H2K$^b$ (OVA-LLC) were kindly provided by K. Zeng, Nanjing University, China. Cells were cultured in DMEM or RPMI 1640 medium containing 10% fetal bovine serum (FBS; Thermo Fisher Scientific, MA), harvested at ~80% confluency, washed twice with PBS, and subcultured for passage. The short tandem repeat (STR) profiling of these cell lines was authenticated (Beijing Microread Genetics Co., Ltd., Beijing, China). Then, all cell lines were detected negative for mycoplasma contamination (Corues Biotechnology, Nanjing, China).

## Separation of primary cytotoxic CD8⁺ T lymphocytes (CTLs)

To extract the CTLs from lung tissue, the lung tissues in the FPM-exposed mice stimulated with LLC or OVA-LLC for indicated days were respectively collected and digested with 2 mg/mL collagenase type I and IV (Thermo Fisher Scientific) for 20 min. Then, a single-cell suspension was prepared using the program m_lung_02.01 on the gentleMACS Dissociator (Miltenyi Biotec, Bergisch Gladbach, Germany). The CTLs were isolated from this single-cell suspension using the CD8⁺ T cell isolation kit with a MidiMACS separator (Miltenyi Biotec).

## Cloning, expression, and purification of PXDN

His-tagged full-length mouse PXDN homolog encoded on indicated vector was generated by GenScript and transfected into HEK 293F cells using Lipo2000 (Invitrogen) according to standard selection and cultivation procedures with minor modifications (*Soudi et al., 2015*). On a large scale, cells were cultivated in Expi293 Met (-) Expression Medium (Thermo Fisher Scientific). The harvested media were stored at 4°C and eventually purified using Ni-NTA Agarose (QIAGEN). Fractions with the best purity number were pooled, concentrated, and desalted using Centricon with a 100 kDa cutoff membrane (Millipore).

## Stimulation of FPM on LLC and CTLs

To analyze the effect of FPM on the proliferation of LLC, cell counting kit-8 (CCK-8) test was performed. Briefly, $1 \times 10^4$ LLCs were seeded in 96-well culture plates and then stimulated with different concentrations of FPM (0, 5, 10, 30, 50, 100, and 500 µg/mL) and cultured for 24 and 48 hr. CCK-8 kit (Dojindo Laboratories, Kumamoto, Japan) was used to examine the proliferation of LLC at indicated time points, with cells treated with 1× PBS as control. To further analyze the effect of FPM on the migration potential of T cells, the CTLs were stimulated with 10 µg/mL FPM, which showed slight cytotoxic, for 48 hr.

## Establishment and treatment of lung tumor model in FPM-exposed mice

### FPM-exposed mice

Mice exposed to FPM were generated according to a previously published method (*Wang et al., 2017*). 6- to 8-week-old C57BL/6J mice or *Foxn1ⁿᵘ* naked mice of the same ground were purchased from Beijing Vital River Laboratory Animal Technology Co. Ltd. (Beijing, China). OT-1 T cell receptor transgenic mice (C57BL/6-Tg [TcraTcrb] 1100Mjb/J) were a gift from K. Zeng (Nanjing University). These mice were randomly divided into three groups (PBS ones, FPM-exposed ones: PM1 and TSP; each group contained at least three mice). Mice were anesthetized by intraperitoneal injection of pentobarbital sodium at 45 mg/kg body weight. After the trachea were exposed by opening the neck skin and blunt dissection, mice received suspension of 0.2 mg FPM in a total volume of 50 µL of sterile physiological saline by inserting a 7-gauge needle (BD Biosciences, San Jose, CA) into the trachea transorally. To be estimated, before intratracheal instillation, FPM suspension was always sonicated and vortexed. After the site of surgery was sutured and cleaned with penicillin, the mice were allowed to recover until they were sacrificed. As a control, PBS was applied in a similar manner. After being exposed to FPM for 7 days, mice were (1) sacrificed for analyzing the changes of lung tissue structure or (2) subsequently stimulated with 5 µg/kg T cell chemokine – C-X-C motif chemokine ligand 10 (CXCL10, PeproTech, Rocky Hill, USA) (*Griffith et al., 2014*), or named as interferon-inducible protein-10 (IP-10) for 2 hr through intratracheal injection, to analyze the CTLs' infiltration into lung tissue, and (3) to analyze the location of PXDN and FPM in the lung, mice were exposed to rhodamine-labeled FPM (R-FPM) with intraperitoneal injection. Lung tissue was extracted for IF 4 hr later.

### Establishment of lung tumor model in FPM-exposed mice

For the syngeneic LLC model, after being exposed to FPM for 7 days, mice were further intravenously injected with $5 \times 10^7$ LLC cells for indicated days (0, 1, 3, 5, 10, and 20 days) to create the lung

carcinoma model post-particle administration. For the LLC-stimulated carcinoma model, 20 days after the injection of LLC, the mice of different groups were sacrificed, and the lung tissues were extracted for analysis. The number of tumors suffered by the mice was examined and evaluated randomly under blindfold conditions. For the transgenic ($Kras^{G12D}Trp53^{-/-}$) mouse models, mice harboring a Cre-inducible endogenous oncogenic $Kras^{LSL-G12D}Trp53^{fl/fl}$ allele (GemPharmatech Co., Ltd., Nanjing, China) were treated as above. 7 days after being exposed to FPM, mice were allowed to inhale $1 \times 10^7$ plaque-forming unit (PFU) Cre adenovirus (AdCre, OBiO Technology [Shanghai] Corp., Ltd., Shanghai, China) to activate $K$-$ras^{G12D}$ expression and knock out p53 in lung tissue ($Kras^{G12D}Trp53^{-/-}$ transgenic model). The mice were sacrificed 50 days after tumor initiation.

## Administration of PXDN inhibitor

A series of plasmids capable of ectopically expressing PXDN-specific short hairpin RNA (PXDN shRNA, shPXDN) or control shRNA (shNC) were designed and constructed by GenePharma Biotechnology, Shanghai, China. For transfection, in vivo-jetPEI (Polyplus Transfection, Illkirch, France) was used as a delivery agent (*Filippi et al., 2009*). The transfection reagent complex (0.16 µL of in vivo-jetPEI per µg plasmid DNA) was prepared and mixed according to the manufacturer's instructions in glucose buffer. Then, 20 µL buffer containing 4 µg shPXDN was delivered into murine lung tissue through trachea injection. Besides, to analyze the effect of small-molecular PXDN inhibitors, the FPM-exposed mice were administrated with 25 mg/kg MMZ (MedChemExpress LLC, Shanghai, China) and 50 mg/kg PHG (Aladdin, Shanghai, China) twice every three days pre- and post-FPM stimulation, respectively.

During the experimental procedure, all animal studies were performed under protocols approved by institutional guidelines (Nanjing University Institutional Animal Care and Use Committee). They were also required to conform to the Guidelines for the Care and Use of Laboratory Animals published by the National Institutes of Health. The mice were housed five per cage and fed in a specific pathogen-free (SPF) animal facility with controlled light (12 hr light/dark cycles), temperature, and humidity, with food and water available.

## Analysis of T cell migration on lung tissue

### Lung tissue slice preparation

To analyze T cell migration on lung tissue, the lung samples of different groups, including the FPM-exposed mice and the FPM-exposed mice pre-treated with PXDN shRNA (shPXDN), were respectively prepared as the 50 µm frozen section (*Salmon et al., 2012*). In some experiments, tissue sections were pretreated with 50 µg/mL collagenase D (Worthington Biochemical Corp., CO) in RPMI 1640 for 5 min, then rinsed in complete RPMI 1640 medium.

### Cell preparation

Jurkat T cells were stained with Calcein-AM (Dojindo Laboratories) for 30 min at 37°C and then washed with HBSS (Sangon Biotech, Shanghai, China) three times. $1.5 \times 10^5$ T cells totally in 10–20 µL were added at one side of the cut surface of each slice. To ensure cells settle down on the slice, slices with T cells were incubated for 1 hr at 37°C, 5% $CO_2$, gently washed to remove the residual cells that had not entered the tissue, and kept at 37°C, 5% $CO_2$ before imaging.

### Time-lapse imaging and cell trajectory analysis

For imaging T cells' migration on the lung tissue slice, 5 µg/mL IP-10 were added on the other side of the slice and images were then acquired in time-lapse model with a SP5 confocal system (Leica) every 3 min for 1 hr. Imaging was exported and compressed into videos in .AVI format. To quantify T cell trajectories with the surrounding ECM in lung tissue, the cell migration video including image sequence cell migration was analyzed with TimTaxis Software (https://www.wimasis.com/en/

WimTaxis) by identification of the centroids of individual cells at consecutive time points. The relationship of statistical data composed of displacement, distance, velocity, and acceleration vs. time was respectively further analyzed.

### Extraction of soluble collagen IV

To generate soluble collagen IV, the fused mouse bone marrow fibroblast M2-10B4 cells were plated at high density and maintained at confluency for 7 days in the presence of 50 µg/mL ascorbic acid (Sangon Biotech), with media changes every 24–36 hr. Crosslinking was inhibited by supplementing the culture conditions with indicated concentrations (0, 50, 100, 200, 300, and 500 µM) of PHG. PHG and ascorbic acid treatments were initiated upon confluency. With the 200 µM PHG, which could be sufficient to inhibit the Col IV crosslink, after the M2-10B4 cells were stimulated for 7 days and collected through scrape, cultured cells and matrix were homogenized in 1% (w/v) deoxycholate (Aladdin) with sonication, and the insoluble material isolated after centrifugation at 20,000 × $g$ for 15 min. Then, the pellet was lysed with RPMI (Beyotime Biotechnology, Shanghai, China) in ice for 30 min. The supernatant containing soluble Col IV was collected after centrifugation at 20,000 × $g$ for 10 min and then incubated with 1 mg/mL FPM per se, 1 mL LH or the mixture of FPM-LH (with the volume ratio of 1:10) for 4 hr. Then, the samples were collected for further WB analysis to detect the change of Col IV crosslink. Besides, to perform cellular experimental analysis, the M2-10B4 cells were incubated with 200 µM PHG for 24 hr and then treated with the same stimulation for 24 hr. The cell samples were collected for further IF analysis.

### Analysis of crosslinking extent of different collagen

Collagen crosslink was assessed biochemically by separating different collagen fractionation via serial extractions, including neutral salt (freshly secreted collagens and procollagens), acetic acid (more mature collagens), and acid pepsin (fibrillar, moderately crosslinked collagens) and insoluble high-crosslink ones from fresh lung tissue as reported in the literature (*Popov et al., 2011*; *Ikenaga et al., 2017*). Briefly, the whole-lung tissue was homogenized in neutral salt buffer (0.5 M NaCl, 0.05 M Tris, pH 7.5; Sangon Biotech) and incubated at 4°C overnight on a rotary shaker. After centrifugation at 24,000 × $g$ for 30 min, the supernatant was collected (fraction A: neutral salt-soluble collagen). The resulting pellet was then extracted with 0.5 M acetic acid (fraction B: acid-soluble collagen; Sangon Biotech), followed by pepsin (2 mg/mL in 0.5 M acetic acid, fraction C: pepsin-soluble collagen; Sangon Biotech). The remaining insoluble fraction D represents mature, highly crosslinked collagen. Then, type I, III, and IV collagens with different extractions were analyzed with corresponding ELISA kits (Nanjing Jiancheng Bioengineering Institute, Nanjing, China). The level of collagen crosslink was calculated as collagen in fraction D divided by total collagen summed by factions A, B, C, and D.

### Detection of crosslinking sites in Col IV
Mass spectrometry and identification of sulfilimine bond crosslinked peptides

To analyze the effect of FPM on crosslink of collagen IV, the potential reaction site containing sulfilimine bond (NC1 domain formed along with C-terminal aggregation) was detected with LC-MS (*Bhave et al., 2012*; *McCall et al., 2014*). Briefly, 5 mg/mL soluble and commercially available collagen IV (Sigma-Aldrich, St. Louis, MO), which was extracted from murine sarcoma basement membrane, was incubated with LH or LH-FPM mixture, 100 µM $H_2O_2$, and 200 µM NaBr for 4 hr at 37°C. After centrifugation at 12,000 rpm for 20 min, the crosslink pellet was collected. Then, the pellet was digested with collagenase D (50 µg/mL; Worthington) for 30 min at 37°C to yield the peptide containing the crosslink site. After centrifugation at 12,000 rpm for 20 min, the supernatant containing the crosslinked peptides was collected. After separation by SDS-PAGE, NC1 domain was digested with trypsin (MS Grade, Thermo Fisher Scientific) overnight at 37°C and then analyzed by LC-MS analysis on a Shimadzu UFLC 20ADXR HPLC system in line with an AB SCIEX 5600 Triple TOF mass spectrometer (AB SCIEX, Framingham, MA). To analyze low-abundance peptides containing crosslinked domain, targeted methods were performed with PeakView software (AB SCIEX) based on raw continuum LC-MS data. Briefly, full-scan spectra of total ion chromatograph (TIC) diagram were acquired, and LC-MS peptide reconstruct with peak finding was provided. According to calculated theoretic mono-isotopic mass of the

sulfilimine (the mass of two hydrogen atoms was subtracted from the sum of the masses for Met93-containing peptide and Lys211-containing peptide), corresponding mass spectrum (about 5030.425, 5046.425, or 5062.425) with different oxidations of methionine ($M_{ox}$) were specifically searched. To display the difference of crosslink site in these groups, the extract ion chromatography (XIC) diagram based on corresponding mass spectrum was shown and compared.

## Detection of allysine at 7S domain crosslinking site

For 7S domain, the detection of primary product allysine could reflect the level of its crosslink. With the reported specific and efficient probes to allysine (*Waghorn et al., 2017*), crosslink of the soluble Col IV incubated with LH or the mixture of LH-FPM was respectively analyzed. Briefly, the 5 mg/mL soluble collagen IV was incubated LH or LH-FPM mixture and 5 mM probes for 30 min at 37°C. Then, the fluorescence intensity was detected with the exciting wavelengths at 565 nm on the microplate reader (Thermo Fisher Scientific). To quantify the yielded allysine in these groups, the indicated oxidized bovine serum albumin (BSA) containing known aldehydes as the standard (oxidized BSA: 16 nM aldehyde/mg; BSA: 1.2 nM aldehyde/mg). For the oxidized BSA, sodium aspartate (13 mg) was added to 50 mg/mL BSA in PBS (2 mL), followed by the addition of a solution of ferric chloride (10 μL, 10 mM) and left to stir at room temperature overnight. A BSA protein standard without the addition of ferric chloride was run in parallel as a control.

## Analysis of PXDN enzyme activity and reaction
### Measurement of PXDN activity

To analyze the effect of FPM on PXDN, the enzyme activity was analyzed with Amplex Red Hydrogen Peroxidase Assay Kit (*Péterfi et al., 2009*) (Thermo Scientific). Briefly, after incubated with FPM for 30 min at 37°C, the enzyme activity was detected in reaction containing 50 mM Amplex Red reagent, 1 mM $H_2O_2$, and PXDN (PeproTech) mixed in FPM. After incubation for 30 min at room temperature, fluorescence was measured at excitation wavelength 590 nm with a fluorescence microplate reader (Thermo Fisher Scientific). To avoid interference of particles per se, the equal FPM was set as negative control.

### Analysis of PXDN enzyme kinetics

To detect the change of PXDN catalytic efficiency after its incubation with FPM, PXDN enzyme kinetic behaviors were investigated according to the Michaelis–Menten model. Briefly, after incubated with FPM for 30 min at 37°C, 10 mU PXDN was mixed with a serial concentration of $H_2O_2$ (0, 0.25, 0.5, 1, 2, 4, 10, 20, 50, 100, and 200 μM). The enzymatic reaction was analyzed by adding 50 mM Amplex Red detection reagent (*Li et al., 2012*). The fluorescent reaction was measured every 40 s for 8 min and subsequently every 5 min six times using a microplate reader (Thermo Scientific) with the enzyme kinetics model. Line weaver–Burk representative plot was generated from the relationship of reaction rate and substrate concentration. Based on the plot, the Michaelis constant (Km) reflecting the binding efficiency of the enzyme with the substrate and turnover number (Kcat) indicating the enzymatic efficiency were calculated.

### Detection of hypobromous acid

Based on the fact that NADH could be stably brominated into NADH bromohydrin after its reaction with hypobromous acid, hypobromous acid generated by PXDN and its bromination activity was tested by TripleTOF 4600 LC-MS/MS according to the reported literature with a little modification (*Bathish et al., 2018*). Briefly, 100 nM PXDN or the mixture of PXDN and FPM was incubated at 37°C in PBS containing 200 μM NADH and 200 μM NaBr for 30 min. Reactions were started upon addition of 100 μM $H_2O_2$. Then, the supernatant was separated and analyzed to detect NADH and its bromohydrin products. NADH was measured using the transition *m/z* 664.2–408.1, and the bromohydrins by *m/z* 760.2 and 762.2 both going to *m/z* 680.2. Intensity of peaks was calculated using PeakView

software (AB SCIEX). Then, the ratio of NADH bromohydrin relative to NADH based on their peak intensity was calculated.

## Measurement of hypochlorous acid

Reactions were initiated with the addition of 100 µM $H_2O_2$ and 200 mM NaCl after 1 µg PXDN was incubated with FPM for 30 min. 5 µM HClO-detecting fluorescent probes (kindly provided by ICMS, University of Macau) were added to react for 30 min, and fluorescence intensity was determined at excitation wavelength 488 nm with a fluorescence microplate reader (*Xing et al., 2018*). Then, HOCl production was calculated according to the standard curve of a serial concentration of HOCl vs. absorbance. To be estimated, HOCl standards should be freshly prepared by adjusting the pH of NaClO to 7.4 to create HOCl solutions before each time.

## Measurement of NC1 crosslink by PXDN

To delineate sulfilimine crosslink of NC1 fragment in the collagen IV, the solubilized NC1 monomer purified from mouse renal basement membrane (Chondrex, Inc, WA, USA) was incubated with PXDN or the mixture of PXDN and FPM. To initiate the reaction, 100 µM $H_2O_2$ and 200 µM NaBr were respectively added. After incubation for 30 min at 37°C, to visualize the change of sulfilimine crosslinked dimeric ($NC1_{di}$) and non-crosslinked monomeric subunits ($NC1_{mo}$), the solution underwent SDS-PAGE under nonreducing conditions followed by Coomassie Blue staining.

## Preparation and separation of FPM's protein corona

First, LH was extracted from the lungs of healthy mice according to the institutional bioethics approval. Briefly, the extracted lung tissue samples were homogenized in equal volumes of 1× PBS by a homogenizer (approximately three mice/mL LH), and then centrifuged to remove the debris to obtain LH. 10 mg/mL FPM were incubated with LH at the volume ratio of 1:10 under stirring at 4°C for the indicated time. Then, the mixture was centrifuged through a 0.3 M sucrose cushion for 20 min at 4°C at 15,300 × *g* in order to separate the nanoparticle-corona complexes. Then, after rinsing with 1× PBS three times, proteins in the corona were eluted by adding RIPM lysis buffer (50 mM Tris pH 7.4, 150 mM NaCl, 1% Triton X-100, 1% deoxycholate, 0.1% SDS) to the pellet on ice for 1 hr. After centrifugation (20 min at 15,300 × *g* at 4°C), the supernatant-enriching protein corona was collected and stored at –20°C.

## LC-MS analysis and database searches of protein corona

The protocol to analyze protein with LC-MS adhered to a method described previously (*Wang et al., 2017*; *Zhang et al., 2015*). Briefly, after samples were quantified with bicinchoninic acid protein assay kit, 100 µg total protein was reduced by adding 1 M DL-dithiothreitol (Sigma-Aldrich; 60°C, 1 hr), and free cysteines were alkylated with 1 M iodoacetamide (Sigma-Aldrich; room temperature, 10 min in the dark). The alkylated proteins were centrifuged in the 10K ultrafiltration tube (Thermo Fisher Scientific), and the proteins were retained in the 10K ultrafiltration tube. The proteins were further washed with 100 mM tetratehylammonium bromide three times at 4°C for 20 min by centrifugation at 12,000 rpm. Then, the protein was digested with 2 µg porcine sequencing grade trypsin (LC-MS Grade, Sigma-Aldrich) overnight at 37°C. After digestion, the resulting peptides were collected (12,000 rpm, 20 min, 4°C), desalted by Zeba Spin Desalting Columns (Thermo Fisher Scientific), and further enriched by C18 reversed-phase columns (Epoch Life Science, Missouri City, TX). The samples were then subjected to LC-MS analysis. To identify the composition of protein corona, identification of peptides and proteins from continuum LC-MS data was performed with the ProteinPilot 4.5 software (AB SCIEX). Proteins were analyzed by searching the mouse taxon of the UniProtKB/SwissProt database (release 2011_11). The proteins with at least one specific high-scoring peptides were detected and exported from ProteinPilot for the final LC-MS data file at the protein level.

## Identification and characterization of PXDN's liquid-liquid phase separation (LLPS)

Microscopy analysis of LLPS

To analyze the droplet formation of PXDN under the stimulation of FPM, 10 mM fluorescein isothiocyanate (FITC)-labeled proteins were incubated with 50 µg/mL rhodamine B-labeled FPM for 30 min in LH. Then, samples of different groups were dropped onto a glass slide and sealed with a coverslip. Phase separation of PXDN and its liquid-like droplets was observed under phase-contrast and confocal microscopy with a 100× Oil objective (Nikon). The distribution profiles of fluorescence intensity of liquid-like PXDN and FPM were respectively analyzed with the Nikon NIS-Elements software. Besides, to predict the domain that might trigger PXDN's accumulation, the intrinsically disordered regions (IDRs), the domain frequently closed to proteins' phase separation, of PXDN is predicted based on the IUPred algorithm (https://iupred2a.elte.hu/).

FRAP assays

After 10 mM FITC-labeled proteins were incubated with 50 µg/mL rhodamine B-labeled PM1 for 30 min in LH, fluorescence recovery after photobleaching (FRAP) experiments were performed on FITC-labeled PXDN droplets formed in PM1. The photorecovery behavior was tracked using the 488 nm laser line of a 40 × 1.0 NA objective on Zeiss LSM 980 with 2.4-fold magnification. Photobleaching was done with 100% laser power to 30% intensity using the bleaching program of the ZEN software, and time-lapse images were recorded every 10 s. After bleaching, the fluorescence intensities were measured and collected by mean ROI (photo-bleached region and control region without bleach). The raw data with three bleach treatments were processed and analyzed with GraphPad Prism.

## Molecular docking on the effect of phase separation on enzymatic reaction

Template crystal structures of PXDN and NC1 domain in Col IV were identified and downloaded from Swiss Model Protein Data Bank as PDB files (PXDN: 5MFA.1A; NC1: 5NAY). Besides, the putative structure under phase transition, which simulated the PXDN assembly, was created through homology modeling based on one experimentally determined structure of PXDN-related family member (PDB ID: 4C1M) in the RCSB Protein Data Bank (*Bernardes et al., 2015*). Subsequently, the most similar template conformation with 49.8% consistency was chosen from among the candidates and named as the PXDN Dimer. Then, ZDock protocol was used for molecular docking analysis of the interaction between PXDN and NC1 (respectively for the 'solution state' to the 'assembly state'). Docking models of the intuitive contacting interface were outputted after scoring and selection. The interactive area was especially labeled.

## Western blotting

According to the standard protocol, different proteins were separated by SDS-PAGE. To be estimated, the proteins in corona from the nanoparticles were eluted with equal and adequate PAGE sample buffer containing 1 mM phenylmethanesulfonyl fluoride (Sigma-Aldrich) and the same volume of eluted corona proteins was analyzed. Besides, to further estimate the content of PXDN adsorbed on the FPM, different amounts of PXDN (50, 100, 500, 1000, and 2000 ng), together with the corona protein samples, were separated by SDS-PAGE and analyzed by WB. Then, the proteins were transferred onto the polyvinylidine difluoride membranes (Bio-Rad, CA). The membranes were blocked with skim milk and then incubated with primary antibody – PXDN (Merck Millipore), type I collagen (Col I, Boster Biological Technology Co. Ltd., Wuhan, China), type III collagen (Col III, ABclonal Technology, Wuhan, China), type IV collagen (Col IV, Abcam, Cambridge, MA), and glyceraldehyde-3-phosphate dehydrogenase (Abcam) at 4°C with gentle shaking overnight. After being washed with PBS with 0.1% Tween-20 five times, the membrane was incubated with horseradish peroxidase-conjugated antirabbit, antimouse, or antigoat IgG (Life Technologies, Grand Island, NY) at room temperature. After rinsing, positive signal was visualized using an enhanced chemiluminescence system (Cell Signaling

Technology). The band intensity was quantitated using ImageJ software (http://rsb.info.nih.gov/ij/), and the statistical analysis of three independent experiments was performed.

## RNA isolation and quantitative real-time PCR

RNA of cells or lung tissues were extracted by using TRIzol reagent (Life Technologies). For mRNA detection, RT-PCR was launched in an ABI 7300 Fast Real-time PCR System (Applied Biosystems, Foster City, CA) using the SYBR Prime Script RT-PCR Kit (Takara Bio, Shiga, Japan). Each sample was analyzed in triplicates and repeated for three or four independent assays with β-actin as internal control. Primers of integrin-1 (ITGB1), C-X-C motif chemokine receptor 3 (CXCR 3), Rho-associated kinase (ROCKi), and PXDN are listed as follows (Shanghai Generay Biotech Co., Ltd., Shanghai, China):

> ITGB1-forward: 5'-CGTGGTTGCCGGAATTGTTC-3'
> ITGB1-reverse: 5'-ACCAGCTTTACGTCCATAGTTTG-3';
> CXCR3-forward: 5'-TACCTTGAGGTTAGTGAACGTCA-3'
> CXCR3-reverse: 5'-CGCTCTCGTTTTCCCCATAATC-3';
> ROCKi-forward: 5'-AACATGCTGCTGGATAAATCTGG-3'
> ROCKi-reverse: 5'-TGTATCACATCGTACCATGCCT-3';
> PXDN-forward: 5'-CCTGTGTTTCCGTACCACCG-3'
> PXDN-reverse: 5'-CTCTGATTCTGTTGAACCGAAGA-3';
> β-actin-forward: 5'-GGCTGTATTCCCCTCCATCG-3'
> β-actin-reverse: 5'-CCAGTTGGTAACAATGCCATGT-3'.

## Flow cytometry analysis

Lung tissues were digested with 2 mg/mL collagenase type I and IV (Thermo Fisher Scientific) for 30 min to generate a single-cell suspension. Cell suspensions were filtered through 70 μm cell strainers, and red blood cells were lysed. For the intracellular staining, $1 \times 10^6$ cells/mL were treated with the cell activation cocktail (BioLegend, San Diego, CA) according to the manufacturer's protocol. After cells were washed with PBS containing 1% BSA, cells were blocked with 1% BSA at 4°C for 30 min. Zombie Violet Fixable Viability Kit was used for live/dead cell determination. Then, cells were stained on ice for 30 min with surface-staining antibodies, FITC anti-mouse CD45, BV711 anti-mouse CD3, APC anti-mouse CD8a, and then washed, fixated, and permeabilized with the fixation/permeabilization solution kiT (BD Biosciences) and stained with cytokine PE anti-mouse interferon gamma (IFN-γ) antibodies in the dark for 30 min at 4°C. The samples were centrifuged at 400–500 × g for 5 min at 4°C to remove unbound antibody. After rinsing three times, each sample was resuspended for analysis using a BD Fluorescence-Activated Cell Sorter (FACS) Calibur (BD Biosciences). Unconjugated antibodies and IgG controls were run in parallel to set the background. All antibodies and their isotype control antibodies were obtained from BioLegend.

## Enzyme-linked immunospot assay (ELISpot)

The lung tissues exposed to PBS or FPM were excised after the intravenous stimulation of LLC-OVA cells for 1 day. After CTLs were separated, $1 \times 10^5$ CTLs were added in each well of IFN-γ antibody precoated plate and stimulated by $4 \times 10^4$ irradiated LLC-OVA cells or phorbol-12-myristate-13-acetate (PMA, the positve control) for 24 hr in RPMI-1640 supplemented with 10% FBS, 100 U/mL penicillin, and 0.1 mg/mL streptomycin. IFN-γ-producing CTLs were enumerated by a mouse IFN-γ precoated ELISpot kit (Dakewe Biotech Co., Ltd.) according to the manufacturer's instructions. The results were analyzed by AID iSpot (AID-Autoimmun Diagnostika GmbH, Strassberg, Germany).

## Immunofluorescence staining

Lung tissue samples were collected, frozen at optimal cutting temperature (OCT) medium (Thermo Fisher Scientific), and cut into sections. The sections or M2-10B4 cells incubated with 10 μg/mL rhodamine-labeled FPM and 1 μg/mL PXDN (at the volume ratio of 1:10) for 1 hr were fixed with 4% paraformaldehyde (PFA, Sigma-Aldrich) and stained with primary antibody at 4°C overnight. The primary antibodies included PXDN, Col I, Col III, Col IV, and CD8. Next, the sections were incubated with secondary antibody Alexa Fluor (Life Technologies) for 1 hr at room temperature, followed by 4',6-diamidino-2-phenylindole (DAPI, Beyotime) for nuclear staining. Then, the sections were imaged

by LSM 980 with Airyscan 2 confocal microscope (Carl Zeiss, Oberkochen, Germany). To further characterize the crosslink level, based on IF mages of Col IV, look-up table (LUT) analysis based on the fluorescence intensity, surface plot analysis based on the invert binary distance of fluorescence distribution, was respectively accomplished using ImageJ (ImageJ Software, National Institutes of Health, Bethesda, MD). Using the 'ridge detection' plugin in ImageJ, binary images of Col IV network were generated, and the related quantitative analysis of junction number and junction density was created and compared. Besides, the EdU-positive percentage was analyzed using the Bioapps Tools in ZEISS ZEN 3.4 (Carl Zeiss).

## Histological studies

The lung tissue fixed in 2.5% PFA was embedded in paraffin and cut into sections for the H&E and Masson's trichrome staining (NanJing KeyGen Biotech Co., Ltd., Nanjing, China) according to the manufacturer's instructions with slight modification. Stained sections were photographed at different magnification under a microscope. Under blindfold conditions with standard light microscopy, tumor burden (based on the percentage of the area of tumor regions versus that of total lung) according to H&E-stained sections of all five lung lobes was quantified with ImageJ software. Besides, to observe the interstitial ECM structure, lung tissues were fixed with glutaraldehyde at 4°C for 48 hr, dehydrated with an ethanol gradient, and dried at the critical point. Then, the samples were sprayed with gold particles and observed with SEM (SFEG Leo 1550, AMO GmbH, Aachen, Germany).

## Statistical analysis

The results are expressed as mean ± standard deviation (SD). Data were statistically analyzed using Prism Software (GraphPad) and assessed for normality or homogeneity of variance. Differences between multiple groups were compared using one-way or two-way ANOVA with Dunnett's tests or, if appropriate, repeated-measures ANOVA test with *post-hoc* Bonferroni correction. Differences between the two groups were evaluated using the two-tailed unpaired Student's *t*-test. A value of $p < 0.05$ was considered significant; n.s. indicates not significant.

## Acknowledgements

We thank Professor Ke Zeng at Nanjing University for kindly providing the OT-1 TCR transgenic mice and OVA-LLC cells. This study was funded by the National Natural Science Foundation of China (31971309, 32001069, 81973273), the Natural Science Foundation of Jiangsu Province (BK20200318), and the Fundamental Research Funds for the Central Universities (020814380115). CW acknowledges the financial support from the Science and Technology Development Fund, Macao SAR (FDCT 0018/2019/AFJ, 0060/2020/AGJ), and the University of Macau Research Committee (MYRG2020-00084-ICMS). This study was also supported by the funds from the International Cooperation and Exchange of the Natural Science Foundation of China and the Science and Technology Development Fund (31961160701).

## Additional information

### Funding

| Funder | Grant reference number | Author |
|---|---|---|
| National Natural Science Foundation of China | 31971309 | Lei Dong |
| National Natural Science Foundation of China | 32001069 | Zhenzhen Wang |
| National Natural Science Foundation of China | 81973273 | Junfeng Zhang |
| Natural Science Foundation of Jiangsu Province | BK20200318 | Zhenzhen Wang |

| Funder | Grant reference number | Author |
|--------|------------------------|--------|
| Fundo para o Desenvolvimento das Ciências e da Tecnologia | FDCT 0018/2019/AFJ 0060/2020/AGJ | Chunming Wang |
| Universidade de Macau | MYRG2020-00084-ICMS | Chunming Wang |
| National Natural Science Foundation of China | the funds for the International Cooperation and Exchange 31961160701 | Lei Dong |
| Nanjing University | the Fundamental Research Funds for the Central Universities 020814380115 | Lei Dong |

The funders had no role in study design, data collection and interpretation, or the decision to submit the work for publication.

## Author contributions

Zhenzhen Wang, Conceptualization, Data curation, Formal analysis, Funding acquisition, Project administration, Writing - original draft, Writing - review and editing; Ziyu Zhai, Xuejiao Tian, Data curation, Formal analysis; Chunyu Chen, Zhen Xing, Data curation; Panfei Xing, Methodology; Yushun Yang, Methodology, Visualization; Junfeng Zhang, Conceptualization, Funding acquisition, Writing - review and editing; Chunming Wang, Conceptualization, Funding acquisition, Writing - original draft, Writing - review and editing; Lei Dong, Conceptualization, Funding acquisition, Supervision, Writing - original draft, Writing - review and editing

## Author ORCIDs

Zhenzhen Wang (ID) http://orcid.org/0000-0003-4645-0958
Chunming Wang (ID) http://orcid.org/0000-0001-9185-9678
Lei Dong (ID) https://orcid.org/0000-0002-2013-4191

## Ethics

This study was performed in strict accordance with the recommendations in the Guide for the Care and Use of Laboratory Animals of the National Institutes of Health. All of the animals were handled according to approved institutional animal care and use committee (IACUC) protocols (#08-133) of Nanjing University. The protocol was approved by the Animal Ethical and Welfare Committee of Nanjing University (Permit Number: 2008011). All surgery was performed under sodium pentobarbital anesthesia, and every effort was made to minimize suffering.

## Decision letter and Author response

Decision letter https://doi.org/10.7554/eLife.75345.sa1
Author response https://doi.org/10.7554/eLife.75345.sa2

# Additional files

## Supplementary files

Transparent reporting form

Source data 1. Collected raw data for gels and blots.

## Data availability

All data generated or analysed during this study are included in the manuscript and Supplementary files; Source Data files have been provided for Figures 1, 2, 3, 4 and 5.

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
