## [Editor Report]

This article focused on the bioactivity of inhaled fine particulate matter (FPM) in promoting lung tumor progression. The authors presented carefully performed work with impressive quantity. They shed light on that FPM-accelerated tumorigenesis through disordering interstitial extracellular matrix in lung tissue and subsequently impairing early immune defense to tumor cells. Besides, they found that FPM's bioactivities are endowed by an unexpected enzyme, peroxidasin, related to the collagen crosslink, and the latter's abnormal high enzymatic activity. These findings are promising and provide a new potential target for preventing FPM-relevant diseases.

---

## [Decision Letter]

**Decision letter after peer review:**

Thank you for submitting your article "Air Pollution Particles Hijack Peroxidasin to Disrupt Immunosurveillance and Promote Lung Cancer" for consideration by *eLife*. Your article has been reviewed by 3 peer reviewers, and the evaluation has been overseen by Paul Noble as the Senior Editor and Reviewing Editor. The following individual involved in the review of your submission has agreed to reveal their identity: Zhe-Sheng Chen (Reviewer #2).

The reviewers were positive about your work. If you could address the points raised by reviewer #3 in the discussion it would be appreciated. The reviews are below for your consideration.

*Reviewer #1 (Recommendations for the authors):*

I would say this is a most interesting paper with novel thinking. It is a big ambitious hypothesis, with many interwoven issues and steps, and a single paper would find it hard to conclusively prove such a claim. I would therefore suggest that the paper be published because it is sufficiently well argued to make the idea credible, but would also suggest to the authors to couch their language in a more cautious manner. I know this is sometimes hard to do when one is being reviewed, but maybe the Editor would allow some possibility for the authors to self critique from a secure basis. Personally, I think the article will come across much better, and having confidence in the ideas, I would like to see that.

I think everyone concerned will be well aware of the impossibility of conclusive arguments based on the limited scale of study and examples provided in Figure1,2. In my view the data seems sufficiently strong to somewhat support the claims, but this will always be questioned.

Obviously, part of Figure 3 really purports to establish that the cross linking is catalysed by this 'corona'. [1] For obvious reasons I would be anxious to make absolutely sure of this effect, for once that is done (as it can be shown in simple physiochemical manners), the rest seemed strongly likely. The in vitro mechanistic verification is particularly important in my view because it excludes many of the other cellular effects that themselves could also lead to aberrant matrix, and if it could be made conclusive would I think be itself of great interest. It belongs to a small but growing group of claims in which coronas adsorbed from the environment are catalytic [2] rather than simply transform the cellular and higher level interactions [3]. So, I would ask the authors if they can envisage a really more conclusive experiment for this effect? Can they make a quasi-quantitative experiment in which the evolution of the reaction can be followed? Can they poison the peroxidasin most convincing in these experiments, and see what happens? I think it is possible.

I would also comment, though this could be more of a discussion point, that the cross-linking hypothesis should be affected by different kinds of bare particle surfaces since their propensity to adsorb will change with the original surface. I am not sure this study is really necessary in such a hypothesis building paper, but certainly if I were the authors, I would want to have a series of particles with different adsorbed species and propensities and the levels of peroxidasin correlated with the effects. Again, not a trivial experiment do well, and be sure the original bare surface is stable in situ etc, but feasible.

My feeling is that this should be published, possibly with some consideration of points made by reviewers, but it looks really important for the community to see.

[1] Nat. Nanotechnol. 16, 229-242 (2021).

[2] Nat Catal 4, 607-614 (2021).

*Reviewer #2 (Recommendations for the authors):*

1) Why did the authors focus on PM1 rather than PM2.5 or PM10? How was the administration dose of FPM selected?

2) For Figure 1I and 1J, to further validate the crucial role of CTLs' insufficiency induced by FPM in lung tumorigenesis, the effect of the adoptively transferred CTLs on the tumor burden of FPM-exposed nude mice should be detected.

3) Also, to testify an immune response closely mediates the FPM's pro-tumorigenesis activity, the EdU staining in the lung tissue of nude mice should be supplemented to exclude the direct role of FPM on tumor cells.

4) Figure 4 showed that FPM increased peroxidasin (PXDN) activity. Other than upregulating the enzymatic activity, does the FPM affect PXDN's expression in the lung tissue?

5) In Figure 1F, 5C and 5D, flow cytometry was performed to analyze the CTLs in lung tissue one day or indicated time after LLC stimulation. The authors should discuss whether the FPM exposure also induced the change of other immune cells, especially NK.

6) This study mainly detects the effect of FPM exposure on extracellular matrix and immune response during the early stage of tumorigenesis. The long-term impact of the inhaled particles on lung tumors should also be discussed.

7) For Materials and methods, more details in the section of 'Molecular Docking on The Effect of Phase Separation on Enzymatic Reaction.' should be provided.

*Reviewer #3 (Recommendations for the authors):*

Overall, the hypothesis was tested carefully with both in vitro and in vivo experiments, and the data was convincing. I only have a few comments:

1. A discussion of the hypothesis with patient data would further strengthen the arguments. The tobacco smoke induces a unique mutational signature in lung cancer tissue (a C>A transversion, or signature 4 as reviewed in Helleday et al., 2014, Nature Review Genetics). It would be interesting to test whether the presence of smoke mutational signature (or the smoke history of patients) is associated with decreased infiltration of T cells or an immunosuppressive microenvironment in lung cancer patients. Such association may be tested using processed data in TCGA datasets of lung cancer patients. The mutational signatures are available in Alexandrove et al., 2013 and the immune cell fractions in tumors are available in Thorsson V et al., 2018.

References:

Helleday, T., Eshtad, S. and Nik-Zainal, S. Mechanisms underlying mutational signatures in human cancers. Nat Rev Genet 15, 585-598, doi:10.1038/nrg3729 (2014).

Alexandrov, L. B. et al. Signatures of mutational processes in human cancer. Nature 500, 415-421, doi:10.1038/nature12477 (2013).

Thorsson, V. et al. The Immune Landscape of Cancer. Immunity 48, 812-830 e814, doi:10.1016/j.immuni.2018.03.023 (2018).

2. While it is convincing that the PXDN plays a role in mediating the FPM effects on collagen crosslinking, it is unclear whether PXDN is the only enzyme in this process. Many other proteins were detected in the "protein corona". Is it possible to estimate the relative abundance of PXDN on FPM? If not, it may be explained in the discussion for non-expert readers.

3. It is a nice piece of interdisciplinary work. Some details may be added for readers in different fields. For example, it is difficult to understand why a conformation change of the enzyme is referred to as "phase separation".

4. In Figure 4D, what does the purple color represent in the merged graph?

5. In Supplementary Table S2, a description of the column title, such as "%Cov", is needed.

---

## [Author Response]

Essential revisions:Reviewer #1 (Recommendations for the authors):I would say this is a most interesting paper with novel thinking. It is a big ambitious hypothesis, with many interwoven issues and steps, and a single paper would find it hard to conclusively prove such a claim. I would therefore suggest that the paper be published because it is sufficiently well argued to make the idea credible, but would also suggest to the authors to couch their language in a more cautious manner. I know this is sometimes hard to do when one is being reviewed, but maybe the Editor would allow some possibility for the authors to self critique from a secure basis. Personally, I think the article will come across much better, and having confidence in the ideas, I would like to see that.

Thanks for your suggestion. We have modified our description more cautiously, especially for the sentences related to FPM’s pro-tumorigenesis potential and PXDN’s phase separation.

Obviously, part of Figure 3 really purports to establish that the cross linking is catalysed by this 'corona'. [1] For obvious reasons I would be anxious to make absolutely sure of this effect, for once that is done (as it can be shown in simple physiochemical manners), the rest seemed strongly likely. The in vitro mechanistic verification is particularly important in my view because it excludes many of the other cellular effects that themselves could also lead to aberrant matrix, and if it could be made conclusive would I think be itself of great interest. It belongs to a small but growing group of claims in which coronas adsorbed from the environment are catalytic [2] rather than simply transform the cellular and higher level interactions [3]. So, I would ask the authors if they can envisage a really more conclusive experiment for this effect? Can they make a quasi-quantitative experiment in which the evolution of the reaction can be followed? Can they poison the peroxidasin most convincing in these experiments, and see what happens? I think it is possible.

These questions are extraordinarily inspiring and in line with our plan for future study. First, our finding fully supports the catalytic role of coronas. We had speculated that FPM increased collagen crosslinking by upregulating PXDN expression; however, the data showed no significant difference in the level of PXDN! From this observation, we started focusing on the catalytic activity of corona including PXDN adsorbed on FPM and, through hierarchical clustering, found these coronas rich in enzymes. We are glad to know our finding agrees with a growing group of claims that coronas adsorbed from the environment are catalytic (*Nat. Nanotechnol. 2021.16,229.; Nat. Catal.,2021.4,607*.).

Next, as the reviewer pointed out, the in vitro mechanistic validation is important. We also analyzed the FPM-PXDN interaction in vitro by incubating the particles with the purified fulllength native peroxidasin. The data (including those in Figure 4E-4I) indicated that FPM incubation significantly enhanced PXDN’s enzymatic performance. During these experiments, to quantitatively analyze the HOBr production generated by FPM-PXDN’s catalytic process, LC-MS was employed to determine the bromination of NADH (*Arch. Biochem. Biophys. 2018. 646,120; J. Biol. Chem.2015. 290, 1087; Cell. 2014.157,1380*). We believe a more comprehensive study in future into the kinetics of enzyme-substrate reactions, based on dynamic quantification of this pair, would help describe this evolution process. As the reviewer wisely suggests, poisoning peroxidasin in vitro can also be an interesting experiment to perform.

We will most likely use methods shown as below:

1) Because that the generation of crosslinked dimeric (NC1_di_) is directly related to the consumption of un-crosslinked monomeric subunits (NC1_mo_), the enzymatic performance of FPM/PXDN can be investigated with a NC1_mo_-to-NC1_di_ conversion assay, followed by quantitative western blotting. Besides, after FPM and PXDN were incubated for the indicated time, NC1_mo_-to-NC1_di_ transformation curves could be performed to analyze the correlation between the incubation time and the yield of NC1di.

2) To validate the catalytic reaction mediated by the FPM’s surface, an active dimethyl labelling strategy combined with MS could be applied to systematically probe the lysine-proximal microenvironments of the FPM/PXDN complexes. It’s reported that the lysine residues on the free enzyme are more easily accessed and labeled than those adsorbed on materials’ surfaces (*Nat. Catal. 2021.4,607.; Anal. Chem. 2016. 88, 12060.*). Thus, this method could reflect the state of PXDN.

3) For poisoning the peroxidasin in in vitro experiments, perhaps the adsorption of peroxidasin could be competitively inhibited by the addition of some other targeted proteins by utilizing the preference of FPM’s protein corona. If feasible, it inspires us to design the therapeutic method to prevent FPM’s biological risk by modulating its protein corona.

I would also comment, though this could be more of a discussion point, that the cross-linking hypothesis should be affected by different kinds of bare particle surfaces since their propensity to adsorb will change with the original surface. I am not sure this study is really necessary in such a hypothesis building paper, but certainly if I were the authors, I would want to have a series of particles with different adsorbed species and propensities and the levels of peroxidasin correlated with the effects.

We appreciate this suggestion and agree with the reviewer that this is an interesting topic to explore in our future study. In the current study, due to the complexity of the environmentally collected FPM, we mixed an equal proportion of different samples to eliminate the interference of sampling resources and discovered peroxidasin as the biomarker in the corona related to crosslinking. In future, we can study the physicochemical features of individual samples and their absorbed PXDN levels, followed by linking the abundance to eventual tumorigenic activities.

Reviewer #2 (Recommendations for the authors):1) Why did the authors focus on PM1 rather than PM2.5 or PM10? How was the administration dose of FPM selected?

1) According to the reports related to the fractional deposition of particulate matters inhaled into the respiratory tract (*Respirology.2012. 17, 1031.*), the nanosized components with a diameter less than 1 μm (PM1) displayed the highest deposition efficiency in the alveolar and bronchioles region. Besides, these particles are more likely to disturb the encounters when they interact with cells or subcellular components due to their extraordinarily high number concentrations per given mass and increased surface area per unit mass (*Environ. Health. Perspect. 2005. 113*, 823.). Thus, we focused on the effect of PM1 on lung tissue.

2) For the FPM-exposed experimental model, previous studies adopted different methods. The administration dose ranged from 2 mg/kg to 30 mg/kg. We chose an average one at 10 mg/kg and exposed FPM to mice by intratracheal instillation.

2) For Figure 1I and 1J, to further validate the crucial role of CTLs' insufficiency induced by FPM in lung tumorigenesis, the effect of the adoptively transferred CTLs on the tumor burden of FPM-exposed nude mice should be detected.

Although we did not adoptively transfer CTLs into the nude mice, its effect on the tumor burden could be predicted from Figure. 5C-5F. We observed significantly attenuated tumor development along with the recovery of CTLs’ infiltration in the FPM-exposed lung tissue. These outcomes to some extent could testify CTLs’ crucial role in FPM-related lung tumorigenesis.

3) Also, to testify an immune response closely mediates the FPM's pro-tumorigenesis activity, the EdU staining in the lung tissue of nude mice should be supplemented to exclude the direct role of FPM on tumor cells.

This analysis is very interesting for our future study. The similar tumor burden and tumor number of nude mice suggested little difference in cancerous cells’ proliferation after FPM exposure. This could be further validated by the EdU staining in Figure 1E. All these results remaindered us that the immune response might be the main attribution to the FPM's protumorigenesis activity.

4) Figure 4 showed that FPM increased peroxidasin (PXDN) activity. Other than upregulating the enzymatic activity, does the FPM affect PXDN's expression in the lung tissue?

Thanks for this insightful question. Actually, in our preliminary experiment, we had speculated that FPM promoted Col IV crosslink by increasing PXDN’s expression in the lung tissue. However, we analyzed the transcription and protein’s expression level of PXDN through qRT-PCR and WB analysis and found little difference before and after FPM exposure. These interesting results pushed us to focus on the change of PXDN’s enzymatic activity after its adsorption on FPM’s surface. And that’s the main reason why we explored the catalytic analysis of PXDN, rather than its direct cellular biological effect.

5) In Figure 1F, 5C and 5D, flow cytometry was performed to analyze the CTLs in lung tissue one day or indicated time after LLC stimulation. The authors should discuss whether the FPM exposure also induced the change of other immune cells, especially NK.

We thank the reviewer for this excellent question. When T cells crawl in the lung tissue, the extracellular matrix serves as their migration scaffolds (*Nat. Rev. Immunol. 2014. 14, 232*).

Similarly, NK cells use such migration strategies in the lung interstitial space (*Trends. Cell Biol. 2020. 30*, 818; *ELife. 2022. 11, 76269*). Based on our study, migrating CTLs could be repelled by dense ECM bundles induced by FPM. We speculated that the same scenario applies to NK cells. FPM exposure might also impair the NK’s mobility in the lung tissue.

6) This study mainly detects the effect of FPM exposure on extracellular matrix and immune response during the early stage of tumorigenesis. The long-term impact of the inhaled particles on lung tumors should also be discussed.

Indeed, the role of extracellular matrix and immune response on the cancerous cells varies along with the tumor development (*Nat. Commun. 2020. 11, 5120; Front. Mol. Biosci. 2020. 6, 160.*), including the initiation, growth, migration, and others. And vice versus, the ECM and immune cells could be dynamically regulated or educated by the tumor cells (*Nat. Immunol. 2013. 14, 1014; Nature. 2021. 599, 673*), which should be also taken into consideration. Taken together, it is challenging to predict the long-term impact of FPM on lung tumor development.

But it is worth being explored in our further study.

7) For Materials and methods, more details in the section of 'Molecular Docking on The Effect of Phase Separation on Enzymatic Reaction.' should be provided.

We have supplemented some details in the section of 'Molecular Docking on The Effect of Phase Separation on Enzymatic Reaction.' in the Revised Manuscript.

Reviewer #3 (Recommendations for the authors):Overall, the hypothesis was tested carefully with both in vitro and in vivo experiments, and the data was convincing. I only have a few comments:1. A discussion of the hypothesis with patient data would further strengthen the arguments. The tobacco smoke induces a unique mutational signature in lung cancer tissue (a C>A transversion, or signature 4 as reviewed in Helleday et al., 2014, Nature Review Genetics). It would be interesting to test whether the presence of smoke mutational signature (or the smoke history of patients) is associated with decreased infiltration of T cells or an immunosuppressive microenvironment in lung cancer patients. Such association may be tested using processed data in TCGA datasets of lung cancer patients. The mutational signatures are available in Alexandrove et al., 2013 and the immune cell fractions in tumors are available in Thorsson V et al., 2018.

We appreciate this insightful suggestion. The mutational signature based on TCGA analysis and immune cell fractions in lung cancer tissue in the patient data could provide solid support for our study (*Nature. 2013. 499, 214.; Science. 2013. 339, 1546; Nature. 2013. 500, 415; Immunity. 2018. 48, 812; Nat. Rev. Genet. 2014. 15, 585.*). We have discussed this interesting issue in *the Revised Manuscript*.

2. While it is convincing that the PXDN plays a role in mediating the FPM effects on collagen crosslinking, it is unclear whether PXDN is the only enzyme in this process. Many other proteins were detected in the "protein corona". Is it possible to estimate the relative abundance of PXDN on FPM? If not, it may be explained in the discussion for non-expert readers.

1) The relative abundance of PXDN on FPM could be estimated according to its rank in the LC-MS list and the quantitative western blot. For the former one, we observed the PXDN both in PM1 and TSP’s corona (PM1: 378/1488;TSP: 417/1290); for the latter one, as shown in Figure 4A, PXDN in the total protein absorbed on FPM was detected by Western blotting, with a serial content of recombinant protein as the standard control. The data showed that about 100 ng PXDN was tethered to FPM.

2) Indeed, PXDN is not the only one enzyme in the protein corona that might affect the collagen crosslink, which also include MPO, EPO, CAPN1 and others. Some of them even displayed higher abundance in the LC-MS list. But among these proteins, PXDN is the specifically one mediating the formation of sulfilimine bond, the central target of FPM’s bioactivity. The activities of other proteins is hardly detectable, which might owe to their conformation change or misfolding during protein corona’s formation (*J. Am. Chem. Soc. 2013. 135, 17359.; Biomaterials.2021. 265, 120452; Adv Mater. 2019. 31, e1805740.*). Thus, we focused on PXDN, the one endows FPM with pro-crosslink activity, despite its abundance is ordinary.

3. It is a nice piece of interdisciplinary work. Some details may be added for readers in different fields. For example, it is difficult to understand why a conformation change of the enzyme is referred to as "phase separation".

This is a great reminder. We have added the definition and characteristics of proteins’ phase separation in the Revised Manuscript.

4. In Figure 4D, what does the purple color represent in the merged graph?

In the merged photograph, this color represents the nuclear staining DAPI, except for the red rhodamine-labelled PXDN and green FITC-FPM. We have added it in *the Revised*

*Manuscript*.

5. In Supplementary Table S2, a description of the column title, such as "%Cov", is needed.

We have added the description in *the Revised Supplementary Table S2*.

Finally, we sincerely thank the three experts for their valuable comments on our manuscript.